

# Impacts of the Horizontal and Vertical Grids on the Numerical Solutions of the Dynamical Equations. Part I: Nonhydrostatic Inertia-Gravity Modes

Celal S. Konor and David A. Randall

Department of Atmospheric Science, Colorado State University, Fort Collins, Colorado, 80523, USA

*Correspondence to:* Celal S. Konor (csk@atmos.colostate.edu)

**Abstract.** We have used a normal-mode analysis to investigate the impacts of the horizontal and vertical discretizations on

the numerical solutions of the nonhydrostatic anelastic inertia-gravity modes on a midlatitude $f$-plane. The dispersion

equations are derived from the linearized anelastic equations that are discretized on the Z, C, D, CD, (DC), A, E, and B

horizontal grids, and on the L and CP vertical grids. The effects of both horizontal grid spacing and vertical wave number are

analyzed, and the role of nonhydrostatic effects is discussed. We also compare the results of the normal-mode analyses with

numerical solutions obtained by running linearized numerical models based on the various horizontal grids. The sources and

behaviors of the computational modes in the numerical simulations are also examined.

Our normal-mode analyses with the Z, C, D, A, E and B grids generally confirm the conclusions of previous shallow-

water studies for the cyclone resolving scales (with low horizontal wavenumbers). We conclude that for cloud-resolving

resolutions (with high horizontal wavenumbers) the Z and C grids become overall more accurate than for the cyclone-

resolving scales, aided by nonhydrostatic effects.

A companion paper, Part II, discusses the impacts of the discretization on the Rossby modes on a midlatitude $\beta$-plane.

## 1. Introduction

In the discretization of the governing equations of atmosphere models, we try to maintain as many physical properties of the

continuous system as possible. These include mimetic properties, such as selected identities from vector calculus; linear

properties, such as the stability of the discrete systems, discrete representations of the geostrophic and hydrostatic adjustment

processes, and discrete wave dispersion that faithfully imitates the continuous dispersion; and nonlinear properties, such as

conservation of energy and enstrophy.

The purpose of this paper to discuss the maintenance of the linear properties of the finite-differenced nonhydrostatic

equations on selected horizontal and vertical grids, with a particular focus on wave dispersion. Although the linear properties

of a discretized system should not be the only factor in selecting a grid, they must be a key consideration. There have been

numerous published studies on the horizontal discretization of the shallow-water equations (e.g. Winninghoff, 1968;





Arakawa and Lamb, 1977; Mesinger and Arakawa, 1976; Randall, 1994; Skamarock, 2008; Thuburn, 2008; Thuburn et. al., 2009; Weller et al. 2012), and on the vertical discretization of the quasi-hydrostatic equations (e.g. Tokioka, 1978; Arakawa and Moorthi, 1988; Hollingsworth, 1995; Arakawa and Konor, 1996; Konor and Arakawa, 2000). There are also a few published studies of the vertical discretization of the nonhydrostatic equations (e.g. Girard et al., 2014; and Thuburn and

Woolings, 2004; Toy and Randall, 2009). We are not aware of any previous publications on the horizontal discretization of the nonhydrostatic equations.

Arakawa and Winninghoff (Winninghoff, 1968; Arakawa and Lamb, 1977; Arakawa, 1988) defined the A, B, C, D and E grids based on the staggering of the variables. More recently, Randall (1994) and Lin and Rood (1997) have respectively added the Z grid and the CD grid to this list. We also examined a DC grid to aid our analysis of the CD grid. For vertical

discretization, there are only two grids available, namely the Lorenz grid (Lorenz, 1960; hereafter, L grid), and the Charney-Phillips grid (Charney and Phillips, 1953; hereafter CP grid). In this paper, we discuss the discretization of the nonhydrostatic equations on these various horizontal and vertical grids.

Our analysis is based on the three-dimensional anelastic system introduced by Lipps and Hemler (1982) because the curl of its pressure-gradient force vanishes, which gives an important simplicity to our normal-mode analyses based on the

vorticity and divergence equations. The anelastic system also excludes the physically insignificant acoustic waves and the Lamb wave, so that we can focus on the much more important inertia-gravity and Rossby waves. Although the anelastic system has inaccuracies, as pointed out by Davies et al. (2003), Arakawa and Konor (2009), Dukowicz (2013), and Dubos and Voitus (2014), a wide range of nonhydrostatic phenomena are accurately described by the anelastic system. Arakawa and Konor (2009) show that the frequency of the inertia-gravity modes for all horizontal scales, and the frequency of the

middle-latitude Rossby modes for medium and small horizontal scales are among the features that can be well simulated by the anelastic system. Since discretization errors are mostly confined near the shortest resolvable scales, our findings can be directly applied to all nonhydrostatic systems.

The use of a three-dimensional nonhydrostatic system to study the impact of the horizontal discretization on the dispersion of the waves has an important advantage over the two-dimensional shallow-water system, in that it allows us to

directly assess the performance of the discretization as a function of the vertical wavenumber, vertical resolution or vertical grid spacing. With the shallow-water system, only an indirect assessment can be made, based on the values of the Rossby radius of deformation divided by the horizontal grid spacing, as discussed by Arakawa and Lamb (1977), Randall (1994) and others. The results of our anelastic normal-mode analyses are generally consistent with the shallow-water analyses discussed by previous authors for the cyclone resolving scales. Here we present a three-dimensional view of the dispersion of the

modes by including an accurate assessment of the performance of the horizontal grids for the nonhydrostatic cloud-resolving scales.



For large horizontal grid spacings, the results of our analyses are also applicable to the quasi-hydrostatic systems. The normal-mode analyses presented by Arakawa and Konor (2009) show that the frequencies of the medium- and large-scale inertia-gravity waves, with horizontal wave lengths of approximately 60 km and larger, are nearly identical in the anelastic and quasi-hydrostatic systems. The dispersion of mid-latitude Rossby waves is also nearly identical in the two systems, with

the exception of the ultra-long waves. The only major difference is that the quasi-hydrostatic system includes the Lamb wave, while the anelastic system does not. We ignore the Lamb wave in our analyses.

The dispersion of inertia-gravity waves leads to geostrophic and hydrostatic adjustments in the nonhydrostatic system, which is key to the maintenance of the geostrophic and hydrostatic balances, respectively. Thus, the accurate simulation of wave dispersion, i.e., the phase and group velocities are essential to maintain the approximate balances, and also bolster the

stability of the discrete system.

The discretization process can give rise to computational modes. The sources and behaviors of the computational modes differ for each horizontal and vertical grid. While a normal-mode analysis is an excellent method to study the modification of the physical modes with the discretization, it is not completely adequate to study the computational modes. This is one motivation for including in our study analyses of the numerical solutions of the linearized anelastic equations.

This paper discusses the dispersion of the inertial-gravity waves on a midlatitude $f$-plane. Section 2 presents the linearized anelastic equations. Section 3 discusses the horizontal discretization of these equations on the Z, C, D, CD, (DC), A, E, and B grids, in that order. At the end of Sect. 3, we present plots of the discrete dispersion relations, and a comparison of the performance of the grids in simulating the inertia-gravity waves. Vertical discretization on the L and CP grids is discussed in Sect. 4. Section 5 presents a summary table of the discrete dispersion relations, to facilitate comparisons. In

section 6, we present results from simulations of the inertia-gravity modes obtained by running linearized numerical models based on the various horizontal grids, and discuss the sources and behaviors of the various computational modes. Section 7 analyses the nonhydrostatic effects on the performance of the grids by comparing the dispersion of the inertia-gravity modes with the nonhydrostatic, quasi-hydrostatic and shallow-water systems. A summary and conclusions are provided in Sect. 8. Additional details are given in the supplementary information. Finally, Part II (Konor and Randall, 2017) discusses the

dispersion of Rossby modes on a midlatitude $\beta$-plane.

**2. Linearized anelastic equations with an isothermal basic state**

The linearization of the Lipps and Hemler (1983) anelastic equations is with respect to a resting isothermal basic state, following Arakawa and Konor (2009), Davies et al. (2003) and many others. All variables are weighted by the square root of the basic state density $\rho_0^{1/2}(z)$, so that the dispersion relation does not have an imaginary part.





### 2.1 Basic equations

The linearized horizontal momentum equation is

$$\frac{\partial \mathbf{v}}{\partial t} = -f\mathbf{k} \times \mathbf{v} - \nabla_{\mathrm{H}} P \,, \tag{1}$$

where $\mathbf{v}$ is the horizontal velocity, $t$ is time, $f$ is the Coriolis parameter, $\mathbf{k}$ is the vertical unit vector, $\nabla_{\mathrm{H}}$ is the horizontal del operator, $P$ is the pressure. Although the definition of $P$ is not necessary for this study, in the Lipps-Hemler anelastic system, it is given by $P \equiv \rho_0^{1/2} c_p \theta_0 \pi'$, where $c_p$ is the specific heat of dry air under constant pressure, $\theta_0 = \theta_0(z)$ is the basic state potential temperature, $\pi' \equiv \pi - \pi_0$ is the perturbation Exner pressure, $\pi \equiv (p/p_{00})^\kappa$ is the Exner pressure, $p$ is the pressure, $p_{00}$ is a standard pressure, $\kappa \equiv R/c_p$, $R$ is the gas constant, $\pi_0 = \pi_0(z)$ is the basic state Exner pressure,

$\theta \equiv T/\pi$ is the potential temperature and $T$ is the temperature.

By taking $\mathbf{k} \cdot \nabla_{\mathrm{H}} \times (1)$, we obtain the linearized vertical vorticity equation as

$$\frac{\partial \omega_z}{\partial t} = -fD \,, \tag{2}$$

where $\omega_z \equiv \mathbf{k} \cdot \nabla_{\mathrm{H}} \times \mathbf{v}$ is the vertical vorticity and $D \equiv \nabla_{\mathrm{H}} \cdot \mathbf{v}$ is the divergence of the horizontal wind. By applying the divergence operator to Eq. (1), we obtain the linearized divergence equation as

$$\frac{\partial D}{\partial t} = f\omega_z - \nabla_{\mathrm{H}}^2 P \,, \tag{3}$$

where $\nabla_{\mathrm{H}}^2$ is the horizontal Laplacian operator. To predict the horizontal velocity, we can directly integrate (1). An alternative approach is to first predict vorticity and divergence through Eqs. (2) and (3), respectively, and then obtain the horizontal momentum by using horizontal elliptic solvers. To study the linear modes, we will use the vorticity and divergence equations instead of the momentum equation.

The vertical momentum equation is

$$\frac{\partial w}{\partial t} = -\left( \frac{\partial}{\partial z} - \frac{1}{2\rho_0} \frac{\partial \rho_0}{\partial z} \right) P + B \,, \tag{4}$$



where $w$ is the vertical velocity weighted by $\rho_0^{1/2}$ and $B \equiv \rho_0^{1/2} g\theta'/\theta_0$ is the buoyancy, $g$ is the gravitational acceleration, $\theta' \equiv \theta' - \theta_0$ is the perturbation potential temperature. Note that in Eq. (4) $(1/2)(1/\rho_0)(\partial\rho_0/\partial z)$ appears as a result of the weighting of the pressure term by $\rho_0^{1/2}$ with the linearization. The thermodynamic equation is

$$\frac{\partial B}{\partial t} = -N^2 w \ ,$$
(5)

where $N^2 \equiv (g/\theta_0)(\partial\theta_0/\partial z)$ is the square of the Brunt-Väisälä frequency associated with the basic state. Both $(1/\rho_0)(\partial\rho_0/\partial z) \equiv -1/H$ and $N^2 = (g/\theta_0)(\partial\theta_0/\partial z) = g\kappa/H$ are constants for an isothermal atmosphere, where $H \equiv RT_{00}/g$ is the scale height of the isothermal basic state, and $T_{00}$ is the three-dimensionally and temporally constant temperature of the isothermal basic state. The anelastic continuity equation is

$$D + \left( \frac{\partial}{\partial z} + \frac{1}{2\rho_0}\frac{\partial\rho_0}{\partial z} \right) w = 0 \ ,$$
(6)

where $D \equiv \nabla_H \cdot \mathbf{v}$ is the divergence.

Equations (4)–(6) can be combined to a single equation by eliminating $B$ and $w$, giving

$$\left( \frac{\partial^2}{\partial t^2} + N^2 \right) D = \left[ \frac{\partial^2}{\partial z^2} - \left( \frac{1}{2\rho_0}\frac{\partial\rho_0}{\partial z} \right)^2 \right] \frac{\partial P}{\partial t} \ .$$
(7)

The three-dimensional elliptic equation that can be used to diagnose the pressure term can be obtained by eliminating $\partial D/\partial t$ between Eq. (3) and $\partial/\partial t$ of Eq. (6), and using Eq. (4). The result is

$$\nabla_H^2 P + \left[ \frac{\partial^2}{\partial z^2} - \left( \frac{1}{2\rho_0}\frac{\partial\rho_0}{\partial z} \right)^2 \right] P = f\omega_z + \left( \frac{\partial}{\partial z} + \frac{1}{2\rho_0}\frac{\partial\rho_0}{\partial z} \right) B \ .$$
(8)

Although the elliptic equation (8) is not needed for the normal-mode analysis, the numerical integration of the linearized equations requires the use of a discrete version of Eq. (8).

These equations have solutions that are steady ($\partial/\partial t = 0$), balanced, (i.e., $D = 0$), and quasi-static, given by

$$0 = f\omega_z - \nabla_H^2 P$$
(9a)

and



$$0 = -\left(\frac{\partial}{\partial z} - \frac{1}{2\rho_0}\frac{\partial \rho_0}{\partial z}\right)P + B . \qquad (9b)$$

By using Eq. (9a) in Eq. (9b), we find that

$$0 = -f\left(\frac{\partial}{\partial z} - \frac{1}{2\rho_0}\frac{\partial \rho_0}{\partial z}\right)\omega_z + \nabla_{\mathrm{H}}^2 B \qquad (9c)$$

is also satisfied for the steady balanced state. The majority of the discrete systems that we will discuss have steady balanced

5    state solutions corresponding to Eqs. (9a-c). However, we found some exceptions with the CD grid discretization, depending on the time-integration scheme. We discuss this point further later, and give examples of the schemes that do, or do not have steady balanced solutions.

To obtain the dispersion relation for the inertia-gravity modes on a middle-latitude $f$-plane, we seek solutions of Eqs. (2), (3) and (7) with the form

$$\phi\left(x,y,z,t\right) = \mathrm{Re}\left\{\hat{\Phi}e^{i\left(kx+\ell y+mz-\nu t\right)}\right\} , \qquad (10)$$

where $\phi$ represents an arbitrary variable, Re indicates the real part of a complex function, $\hat{\Phi}$ is the complex amplitude, $\underset{\sim}{i} \equiv \sqrt{-1}$ , $k$ , $\ell$ and $m$ are the wavenumbers of waves propagating in $x$-, $y$- and $z$-directions, respectively, and $\nu$ is the frequency of the waves. Since we assume that the upper and lower boundaries are material surfaces (so that $w_S = w_T = 0$ ), we define the vertical wavenumber $m$ as

$$m \equiv \frac{\pi n}{z_T} \text{ for } n = 1, 2, 3, \cdots \qquad (11)$$

where $n$ is the "integer wavenumber," and $z_T$ is the height of the upper boundary. The use of $n$ is very convenient for the study of discretization because $n$ is equal to the number of layers in a numerical model.

## 2.2 Dispersion of inertia-gravity waves on a mid-latitude $f$-plane

By applying Eq. (10) to $\omega_z$, $D$, and $P$ in Eqs. (2), (3) and (7) and seeking nontrivial solutions, we obtain the dispersion

20    relation as





$$v^2 = \frac{N^2\left(k^2 + \ell^2\right) + f^2\left(m^2 + \dfrac{1}{4H^2}\right)}{\left(k^2 + \ell^2\right) + \left(m^2 + \dfrac{1}{4H^2}\right)}. \tag{12}$$

Here we have used $\left(1/\rho_0\right)\left(\partial\rho_0/\partial z\right) \equiv -1/H$ and $N^2 = \left(g/\theta_0\right)\left(\partial\theta_0/\partial z\right) = g\kappa/H$. It should be noted that the dispersion equation (12) is obtained by seeking nontrivial solutions for $D$ among Eqs. (2), (3) and (7). If we seek nontrivial solutions for $\omega_z$ or $P$, we will find an additional mode with $v = 0$, which corresponds to the steady-state balanced solution. This comment applies to all discrete systems with the exception of some on the CD grid. We will briefly discuss the exceptions with the CD grid. According to Eq. (12) the frequency of inertia-gravity modes is bounded between the Brunt-Väisälä frequency ($N$) and the inertial frequency ($f$). For large horizontal and small vertical wave numbers, the "gravitational term," i.e., the first term in the numerator of Eq. (12), dominates the "inertial term." The effect of rotation on these gravity modes is small.

## 3. Horizontal discretization of the linear anelastic equations on different grids and discrete dispersion equations

The modes of the continuous system will be referred to as physical modes or true modes. The discrete solutions contain versions of these physical modes that are modified by discretization, and they may also contain additional computational modes (or solutions) that are purely the result of discretization. In some cases, the computational modes are dynamically inert and confined to the smallest resolvable scales. In other cases, the computational modes appear in the form of non-unique (or multiple) physical solutions that do not interact in the linear system. These non-unique solutions can also be dynamically inert on the smallest resolvable scales. The generation and behavior of computational modes in the horizontally discrete systems will be discussed in detail at the end this section.

We will try to avoid the repetition of results that have been previously presented by other authors. Many additional aspects of the A, B, C, D and E grids are discussed by Arakawa (2000 and the references therein). For additional discussion on the Z grid, the reader is referred to Randall (1994) and Chen (2015). Lin and Rood (1997) and Skamarock (2008) can be consulted for additional discussions on the CD grid. For recent applications of the C-grid, the paper by Weller et al. (2012) and the references therein are recommended.

We use the vorticity and divergence equations in our normal-mode analyses for all grids, since the inertia-gravity modes are primarily controlled by the vorticity, divergence and pressure (or mass), and the horizontal velocity is determined by the vorticity and divergence.





Next we discuss the dispersion and modification of the physical modes in the discrete solutions through a normal-mode analysis similar to the one used for the continuous system. Fig. 1 shows portions of the horizontal grids discussed in this paper.

### 3.1 Solutions on the Z grid

We choose to discuss the Z grid first because, as pointed out by Randall (1994), it yields better solutions than the A, B, C, D and E grids in the discretization of the shallow-water equations on an $f$ -plane. Instead of predicting the horizontal velocity, a model that uses the Z-grid predicts the vorticity and divergence without staggering. This approach requires the use of elliptic solvers to obtain the horizontal velocity from the predicted vorticity and divergence. The elliptic solvers add 10-20% to the computational cost of the dynamical core.

### 3.1.1 Discrete dispersion of the inertia-gravity waves on an *f*-plane

The vertically and temporally continuous and horizontally discrete versions of vorticity and divergence equations given by Eqs. (2) and (3), respectively, and Eq. (7) can be written on the Z-grid shown in Fig. 1a as

$$\frac{\partial \left(\omega_z\right)_{i,j}}{\partial t} = -fD_{i,j} ,\tag{13}$$

$$\frac{\partial D_{i,j}}{\partial t} = f\left(\omega_z\right)_{i,j} - \frac{1}{d^2}\left(P_{i+1,j} + P_{i-1,j} + P_{i,j+1} + P_{i,j-1} - 4P_{i,j}\right),\tag{14}$$

$$\left(\frac{\partial^2}{\partial t^2} + N^2\right)D_{i,j} = \left[\frac{\partial^2}{\partial z^2} - \left(\frac{1}{2\rho_0}\frac{\partial \rho_0}{\partial z}\right)^2\right]\frac{\partial P_{i,j}}{\partial t} .\tag{15}$$

By using

$$\phi_{i,j}\left(z,t\right) = \mathrm{Re}\left\{\hat{\Phi}e^{i\left(kdi+\ell dj+mz-vt\right)}\right\}\tag{16}$$

in Eqs. (13)–(15), and seeking nontrivial solutions, we can obtain the discrete dispersion relation as

$$v^2 = \frac{N^2\left(\xi^2k^2 + \eta^2\ell^2\right) + f^2\left(m^2 + \dfrac{1}{4H^2}\right)}{\left(\xi^2k^2 + \eta^2\ell^2\right) + \left(m^2 + \dfrac{1}{4H^2}\right)} ,\tag{17}$$

where the factors



$$\xi \equiv \frac{\sin\left(\frac{1}{2}kd\right)}{\frac{1}{2}kd} \quad \text{and} \quad \eta \equiv \frac{\sin\left(\frac{1}{2}\ell d\right)}{\frac{1}{2}\ell d} \tag{18}$$

arise from the finite-difference form of the Laplacian of $P$ in the divergence equation (14). As the grid spacing approaches zero ($kd$, $\ell d \to 0$) and therefore $\xi^2 = \eta^2 \to 1$ according to Eq. (18), the discrete dispersion relation Eq. (17) approaches the continuous dispersion equation (12), which confirms that the discrete solution is consistent. This property is shared by all of

the dispersion relations that we discuss in the paper. For the smallest resolvable horizontal scales, for which $kd = \ell d = \pi$, the factors in the Laplacian yield $\xi^2 = \eta^2 = 4/\pi^2 \sim 0.4$. Since these factors are bounded between zero and one, all discrete modes and all variables move or oscillate with a single frequency that is between $N$ and $f$. We conclude that the discrete inertia-gravity modes are purely physical, i.e., they correspond to solutions of the continuous system, although they are slightly distorted by discretization errors.

For each grid, we present plots of the absolute value of the frequency as a function of $k^* \equiv \sqrt{k^2 + \ell^2}$, as given by the dispersion relations obtained for the various schemes for given integer vertical wavenumber, $n$. The true frequency given by Eq. (12) is also superimposed in these plots to facilitate assessment of the results. We use $k = \ell$ to make the plots, which means that the modes would be oriented diagonally on a square horizontal grid. The scale height of the basic state is assumed to be $H = 24\,\text{km}$ (or $N^2 = 1.16 \times 10^{-4}\,s^{-2}$), which corresponds to a typical lower-tropospheric stability. The height of the

domain is $z_T = 80\,\text{km}$, and the Coriolis parameter is $f = 10^{-4}\,s^{-1}$. The smallest vertical grid spacing is given by $\delta z = z_T/n$. We present the frequency plots for the four different horizontal grid spacings $d = 2\,\text{km}$, $d = 10\,\text{km}$, $d = 25\,\text{km}$ and $d = 100\,\text{km}$. A grid spacing of 2 km is selected to represent the cloud-resolving models. The grid spacings of 10 km and 25 km are selected to be representative of mesoscale and global numerical weather prediction models. The grid spacing of 100 km is selected to be representative of climate models. In these plots, we show the frequency lines for ten integer vertical

wavenumbers ranging from $n = 1$ to $n = 1280$, and the corresponding vertical grid spacings are also indicated.

    Figure 2 shows that in the Z-grid solutions, the discrete frequencies associated with small horizontal wavenumbers are virtually identical to the true frequencies for all vertical wavenumbers and horizontal grid spacings. This is expected because the discretization errors must be negligible away from the smallest resolvable horizontal scale (hereafter, SRHS). For horizontal wavenumbers that are near and at the SRHS (indicated by thin dashed vertical lines in the plots), the discrete

frequencies deviate downward from the true frequencies, and the deviations become larger as the vertical scales of the modes become smaller.

    The group velocity, $\partial|\nu|/\partial k^*$, is the velocity with which the wave-energy moves. If the group velocity in the discrete system is substantially slowed down or reversed, relative to that of the continuous system, then wave energy can be





spuriously trapped in the source regions, which may lead to instability. For brevity, we only qualitatively assess the simulated group velocity as implied by the frequency plots. The group velocity of the inertia-gravity waves in the true solutions is always positive (or zero) as shown by the thin lines in Fig. 2. The Z-grid solutions maintain positive (or zero) group velocity, but near the SRHS the Z-grid group velocity is generally smaller than the true group velocity.

These results generally support Randall's (1994) conclusions regarding the Z grid with the low horizontal wavenumbers. Our results suggest that the nonhydrostatic effects reduce the errors for the deep modes with high horizontal wavenumbers. This will be further discussed in Sect. 7.

### 3.2 Solutions on the C grid

The C-grid is arguably the most commonly used grid in atmosphere models. The Mintz-Arakawa UCLA general circulation
model (GCM; Arakawa, 2000), developed starting in about 1965, was the first GCM and possibly the first atmospheric model to use the C-grid (Akio Arakawa, personal communication, 2017). With the C grid, the normal components of the horizontal velocity, i.e., $u$ and $v$, are predicted at the meridional and latitudinal walls, respectively, as shown in Fig. 1b. The mass, vertical velocity and thermodynamic variables are predicted at the cell centers. Following the standard practice, the vorticity and divergence are defined at the corners and centers of the cells on the C grid, respectively (see Fig. 1b). Since
the mass and divergence are predicted at the cell centers and the normal component of velocity is predicted at the cell walls, the C grid is well suited to controlling the divergent component of the velocity. Although the same is not true for the rotational component of the velocity, some improvements have been developed in recent years (e.g., Adcroft et al., 1999; Thuburn et al., 2009; Weller et al., 2012).

### 3.2.1 Discrete dispersion of the inertia-gravity waves on an *f*-plane

We define the discrete vorticity and divergence in terms of the components of the horizontal velocity on the C grid (Fig. 1b) as

$$\left(\omega_z\right)_{i+1/2,j+1/2} \equiv \frac{1}{d}\left(u_{i+1/2,j} - u_{i+1/2,j+1} + v_{i+1,j+1/2} - v_{i,j+1/2}\right),$$  (19a)

and

$$D_{i,j} \equiv \frac{1}{d}\left(u_{i+1/2,j} - u_{i-1/2,j} + v_{i,j+1/2} - v_{i,j-1/2}\right),$$  (19b)

respectively. Then, the vertically and temporally continuous and horizontally discrete versions of the vorticity and divergence equations corresponding to Eqs. (2) and (3) can be written as




$$\frac{\partial(\omega_z)_{i+1/2,j+1/2}}{\partial t} = -f\frac{1}{4}\left(D_{i,j} + D_{i+1,j} + D_{i,j+1} + D_{i+1,j+1}\right),$$ (20)

and

$$\frac{\partial D_{i,j}}{\partial t} = f\frac{1}{4}\left[(\omega_z)_{i+1/2,j+1/2} + (\omega_z)_{i+1/2,j-1/2} + (\omega_z)_{i-1/2,j+1/2} + (\omega_z)_{i-1/2,j-1/2}\right]$$

$$-\frac{1}{d^2}\left(P_{i+1,j} + P_{i-1,j} + P_{i,j+1} + P_{i,j-1} - 4P_{i,j}\right).$$ (21)

By using Eq. (16) and

$$\phi_{i+1/2,j+1/2}(z,t) = \mathrm{Re}\left\{\hat{\Phi}e^{i\left[k\left(i+\frac{1}{2}\right)d + \ell\left(j+\frac{1}{2}\right)d + mz - vt\right]}\right\}$$ (22)

in Eqs. (20), (21) and (15), we obtain the discrete dispersion equation for the inertia-gravity waves on the C-grid as

$$v^2 = \frac{N^2\left(\xi^2 k^2 + \eta^2 \ell^2\right) + \mu^2 f^2\left(m^2 + \dfrac{1}{4H^2}\right)}{\left(\xi^2 k^2 + \eta^2 \ell^2\right) + \left(m^2 + \dfrac{1}{4H^2}\right)},$$ (23)

where $\xi$ and $\eta$ are defined by Eq. (18) and

$$\mu \equiv \cos\left(\tfrac{1}{2}kd\right)\cos\left(\tfrac{1}{2}\ell d\right).$$ (24)

The factor $\mu^2$ appears in Eq. (23) as a result of the averaging of the vorticity and divergence to each other's grid points. As $k$ (and/or $\ell$) approaches $k_{max} = \ell_{max} = \pi/d$ corresponding to the shortest resolvable horizontal scale (hereafter, SRHS), $\mu$ approaches zero, which reduces the effect of rotation. For $k_{max} = \ell_{max} = \pi/d$, so that $\mu = 0$, the rotation term completely vanishes. This means that the interactions between the divergence and vorticity are broken. For this case, the frequency

obtained by Eq. (23) applies only to the divergence and buoyancy, while the frequency associated with the vorticity is arbitrary. We will return to this point in Sect. 6.

The dispersion relations for the C-grid are plotted in Fig. 3. By comparing the four panels of the figure, the first thing we see is that the character of the solutions changes with the horizontal grid spacing. For the $d = 2\text{ km}$ case shown in Fig. 3a, the C-grid frequency solution is almost identical to the Z-grid solution. The solutions differ for the modes with vertical

integer wavenumber $n = 1947$, for which the C-grid solutions generate negative group velocities ($\partial|v|/\partial k^* < 0$). This means that the nonhydrostatic C-grid models using a horizontal grid spacing of 2 km, and a vertical grid spacing of



$\delta z = 41\,\mathrm{m}$ or larger (with a domain depth of $z_T = 80\,\mathrm{km}$) produce wave solutions nearly as accurate as those of the Z-grid. There is one issue with the C-grid that applies for all horizontal grid spacings: The vorticity and divergence decouple at the SRHS (indicated with double dashed lines in the figures) because this mode cannot recognize the effects of rotation. Figs. 3b, c and d show that for $d = 10\,\mathrm{km}$, $d = 25\,\mathrm{km}$, and $d = 100\,\mathrm{km}$ the group velocities are reversed for modes shallower

than $n = 390$ ($\delta z = 205\,\mathrm{m}$) for 10-km horizontal grid spacing, $n = 156$ ($\delta z = 512\,\mathrm{m}$) for 25-km case and $n = 39$ ($\delta z = 2051\,\mathrm{m}$) for 100-km horizontal grid spacing.

As the horizontal grid spacing of a C-grid model decreases, the solution approaches the true solution for an increasingly wide range of vertical scales. This is because the effects of rotation are unimportant for the shortest modes on sufficiently fine horizontal grids. As a result, the effect of the averaging factor $\mu^2$ becomes negligible in the inertial term, and therefore

the difference between the C- and Z-grid physical solutions virtually disappears. As the horizontal scale of the modes approaches the SRHS, the coupling of vorticity and divergence becomes weaker due to the averaging, and finally disappears at the SRHS. We classify the mode with the SRHS as dynamically inert.

These results generally support conclusions of previous studies of the behavior of the C grid for the low horizontal wavenumbers, (Arakawa and Lamb, 1977; Randall, 1994). However, the nonhydrostatic effects on deep modes with small

horizontal scales contribute to the reduction of the errors with the C grid, as with the Z grid. The reasons for this are discussed in Sect. 7.

**3.3 Solutions on the D grid**

We discuss the D grid here in preparation for a discussion of the CD grid. The D-grid was originally proposed for the convenience of maintaining the geostrophic balance between the wind and mass fields. The Mintz-Arakawa model

developed around 1963 used the D-grid (Akio Arakawa, private communication, 2017), but we are not aware of any other atmospheric model that has used it since then. On the D grid, the longitudinal component of velocity $u$ and the meridional component $\upsilon$ are paired with the meridional and longitudinal finite-difference derivatives of $P$, respectively (Fig. 1c). However, the placement of the velocity components is consistent with the placement of the divergence at the corners of the grid cells. This leads to the most unattractive and problematic feature of the D grid: The mass and/or divergence has to be

averaged in the horizontally discrete continuity equation.

On the D grid, the vertical velocity can be placed either at the centers with the mass and thermodynamic variables or at the corners with the divergence. In this paper, we primarily discuss the former option, in which the vertical velocity is placed at the centers, but we discuss the latter at the end of this subsection.



### 3.3.1 Discrete dispersion of the inertia-gravity waves on an *f*-plane

We define the discrete vorticity and divergence from the components of the horizontal velocity on the D grid (Fig. 1c) as

$$\left(\omega_z\right)_{i,j} \equiv \frac{1}{d}\left(\upsilon_{i-1/2,j} - \upsilon_{i+1/2,j} + u_{i,j-1/2} - u_{i,j+1/2}\right),\tag{25a}$$

and

$$D_{i+1/2,j+1/2} \equiv \frac{1}{d}\left(u_{i+1,j+1/2} - u_{i,j+1/2} + \upsilon_{i+1/2,j+1} - \upsilon_{i+1/2,j}\right),\tag{25b}$$

respectively. The vertically and temporally continuous and horizontally discrete versions of vorticity and divergence equations that are respectively given by Eqs. (2) and (3) can be written as

$$\frac{\partial\left(\omega_z\right)_{i,j}}{\partial t} = -f\frac{1}{4}\left(D_{i-1/2,j-1/2} + D_{i+1/2,j-1/2} + D_{i-1/2,j+1/2} + D_{i+1/2,j+1/2}\right),\tag{26}$$

and

$$\frac{\partial}{\partial t}D_{i+1/2,j+1/2} = f\frac{1}{4}\left[\left(\omega_z\right)_{i+1,j+1} + \left(\omega_z\right)_{i,j+1} + \left(\omega_z\right)_{i+1,j} + \left(\omega_z\right)_{i,j}\right]$$

$$-\frac{1}{d^2}\left(\overline{P}_{i+3/2,j+1/2} + \overline{P}_{i+1/2,j-1/2} + \overline{P}_{i+1/2,j+3/2} + \overline{P}_{i-1/2,j+1/2} - 4\overline{P}_{i+1/2,j+1/2}\right),\tag{27}$$

where $\overline{P}_{i+1/2,j+1/2} \equiv \frac{1}{4}\left(P_{i,j} + P_{i+1,j} + P_{i,j+1} + P_{i+1,j+1}\right)$. The discrete version of Eq. (7) is

$$\left(\frac{\partial^2}{\partial t^2} + N^2\right)\left[\frac{1}{4}\left(D_{i-1/2,j-1/2} + D_{i+1/2,j-1/2} + D_{i-1/2,j+1/2} + D_{i+1/2,j+1/2}\right)\right] - \left[\frac{\partial^2}{\partial z^2} - \left(\frac{1}{2\rho_0}\frac{\partial\rho_0}{\partial z}\right)^2\right]\frac{\partial}{\partial t}P_{i,j} = 0 .\tag{28}$$

Using Eqs. (16) and (22) in Eqs. (26)–(28), we obtain the discrete dispersion relation for the inertia-gravity modes on the D grid as

$$\nu^2 = \frac{\mu^2 N^2\left(\xi^2 k^2 + \eta^2 \ell^2\right) + \mu^2 f^2\left(m^2 + \dfrac{1}{4H^2}\right)}{\mu^2\left(\xi^2 k^2 + \eta^2 \ell^2\right) + \left(m^2 + \dfrac{1}{4H^2}\right)},\tag{29}$$

where $\xi$ and $\eta$ are defined by Eq. (18), and $\mu$ is defined by Eq. (24). As with the C grid, the inertial term is multiplied by





the averaging factor $\mu^2$ in Eq. (29). The Laplacian terms are multiplied by $\mu^2$ because the discrete Laplacian in Eq. (27) involves averaging. As discussed below, the multiplication of the gravitational and inertial terms by $\mu^2$ in the numerator of Eq. (29) creates a stationary mode at the SRHS.

The frequency plots for the D grid are shown in Fig. 4. Since the inertia-gravity solutions are qualitatively similar for all
horizontal grid spacings (unlike the solutions on the C grid), the following discussion applies to all grid spacings. The D-grid solutions for all vertical scales converge to a dynamically inert solution in all variables at the SRHS, while the true solutions are oscillating modes (as indicated by the double dashed lines in Fig. 4). As mentioned above, the zero frequency at the SRHS is a result of the factor $\mu^2$ in the numerator of Eq. (29). Since all variables have zero frequency, with or without rotation, the impact of the dynamically inert modes can be more severe than with the C grid. As a consequence, the group
velocities of the D-grid solutions are badly reversed near the SRHS.

These results are in agreement with the conclusions of Arakawa and Lamb (1977) and Randall (1994) for the D grid.

The configuration of the D grid for which the vertical velocity is defined at the corners, where the divergence is predicted, yields a dispersion relation with a numerator identical to that of Eq. (29), but without $\mu^2$ in the denominator. The frequency plots for this case are qualitatively similar to those shown in Fig. 4; they are omitted here for brevity. A detailed
discussion of this version of the D grid, including the corresponding frequency plots, can be found in the supplementary material.

### 3.4 Solutions on the CD grid

The CD grid proposed by by Lin and Rood (1997) was first used in a flux-form semi-Lagrangian dynamical core. It is now also used in GFDL's finite-volume (FV3) dynamical core, and in the NCAR Community Atmosphere Model (CAM). FV3
will also be used in the next-generation numerical prediction model of the U.S. National Centers for Environmental Prediction. The CD grid is built on a predictor-corrector time-integration scheme, in which the C and D grid discretizations are used for the predictor and corrector steps, respectively (see Fig. 1d). Because of this, an analysis of the dispersion properties of the CD grid has to include temporal discretization, as well as spatial discretization.

Lin and Rood (1997) asserted that the CD grid combines the C-grid's superiority for the prediction of the divergent part
of velocity with the D-grid's superiority for the prediction of the rotational part. They claimed that the solutions of the CD grid should be close to those of the Z grid. Skamarock (2008) examined the normal modes of the linearized shallow-water equations discretized on the CD-grid, and concluded that the linear properties of the CD grid resemble those of the D-grid rather than the Z or C grids.





In this section, we examine the performance of the CD grid as inferred from a normal-mode analysis. Guided by the two papers mentioned above, we temporally and horizontally discretize the anelastic equations on the CD grid. For the predictor step, we use the horizontally discrete vorticity and divergence equations on the C-grid, as given by Eqs. (20) and (21), respectively. Then we horizontally discretize the vertical momentum equation (4), the buoyancy equation (5) and the

continuity equation (6). For the corrector step, we use the horizontally discrete vorticity and divergence equations on the D-grid, as given by Eqs. (26) and (27), respectively. We also horizontally discretize the vertical momentum equation (4), and the buoyancy equation (5) and continuity equation (6) on the D-grid. For the predictor and corrector steps, the temporal discretizations can be summarized in compact form as

$$\phi^{(*)} = \phi^{(n)} + \tfrac{1}{2}\tau F^{(n)}, \tag{30a}$$

and

$$\phi^{(n+1)} = \phi^{(n)} + \tau G^{(*)}, \tag{30b}$$

respectively. In Eqs. (30a) and (30b), $\phi$ is an arbitrary variable, the superscripts $(n)$ and $(n+1)$ denote the time steps (the notational conflict with the vertical wave number should not be confusing), $(*)$ denotes a provisional value, $\tau$ is the time step, and $F$ and $G$ represent the tendency terms. Equation (30a) indicates that the provisional values are obtained by

advancing a half-a-time step. All variables denoted by $(n)$ and $(n+1)$ are defined on the D grid, and those denoted by $(*)$ are defined on the C grid. If Eq. (30a) is applied to the vorticity and divergence equations on the C grid, the initial values with superscript $(n)$ need to be averaged to the C grid from the D grid. It should be noted that, on the D-grid, the provisional values of vorticity (and divergence) do not have to be averaged for the use in the tendency term of the divergence (and vorticity) equations. We seek solutions of these equations in the following forms:

$$\phi_{i,j}^{(*)}(z) \equiv \mathrm{Re}\left\{\hat{\Phi}^{(*)}e^{i(kdi+\ell dj+mz)}\right\} \quad \text{and} \quad \phi_{i+1/2,j+1/2}^{(*)}(z) = \mathrm{Re}\left\{\hat{\Phi}^{(*)}e^{i\left[k(i+1/2)d+\ell(j+1/2)d+mz\right]}\right\}, \tag{31a}$$

and

$$\phi_{i,j}^{(n)}(z) \equiv \mathrm{Re}\left\{\hat{\Phi}^{(n)}e^{i(kdi+\ell dj+mz)}\right\} \quad \text{and} \quad \phi_{i+1/2,j+1/2}^{(n)}(z) = \mathrm{Re}\left\{\hat{\Phi}^{(n)}e^{i\left[k(i+1/2)d+\ell(j+1/2)d+mz\right]}\right\}, \tag{31b}$$

where $(n)$ is the time-step counter, and $(*)$ denotes the provisional value of the complex amplitude of the variables after half-a-time-step of integration in the prediction step on the C-grid.

Skipping a few derivation steps (see the supplemental material), we can write the discrete equations that predict the complex amplitudes as follows:

Predictor step on the C-grid:                                  Corrector step on the D-grid:



$$\hat{\omega}_z^{(*)} = \mu\hat{\omega}_z^{(n)} - \tfrac{1}{2}\tau f\hat{D}^{(n)} , \qquad (32)$$

$$\hat{\omega}_z^{(n+1)} = \hat{\omega}_z^{(n)} - \tau f\hat{D}^{(*)} , \qquad (38)$$

$$\hat{D}^{(*)} = \mu\hat{D}^{(n)} + \tfrac{1}{2}\tau\left(f\hat{\omega}_z^{(n)} + L^2\hat{P}_C\right), \qquad (33)$$

$$\hat{D}^{(n+1)} = \hat{D}^{(n)} + \tau\left(f\hat{\omega}_z^{(*)} + \mu L^2\hat{P}_D\right), \qquad (39)$$

$$\hat{w}^{(*)} = \hat{w}^{(n)} + \tfrac{1}{2}\tau\left[-\left(\underset{\sim}{i}m + \frac{1}{2H}\right)\hat{P}_C + \hat{B}^{(n)}\right], \qquad (34)$$

$$\hat{w}^{(n+1)} = \hat{w}^{(n)} + \tau\left[-\left(\underset{\sim}{i}m + \frac{1}{2H}\right)\hat{P}_D + \hat{B}^{(*)}\right], \qquad (40)$$

$$\hat{B}^{(*)} = \hat{B}^{(n)} - \tfrac{1}{2}\tau N^2\hat{w}^{(n)} , \qquad (35)$$

$$\hat{B}^{(n+1)} = \hat{B}^{(n)} - \tau N^2\hat{w}^{(*)} , \qquad (41)$$

$$\hat{D}^{(*)} + \left(\underset{\sim}{i}m - \frac{1}{2H}\right)\hat{w}^{(*)} = 0 , \qquad (36)$$

$$\mu\hat{D}^{(n+1)} + \left(\underset{\sim}{i}m - \frac{1}{2H}\right)\hat{w}^{(n+1)} = 0 , \qquad (42)$$

In (33), $L^2 \equiv \xi^2 k^2 + \eta^2 \ell^2$, (37)

where $\xi$ and $\eta$ are given by Eq. (18), $\mu$ is given by Eq. (24).

In Eqs. (32) and (33), the averaging factor $\mu$ appears as a result of averaging of the vorticity and divergence from the D

grid to the C grid. The provisionally predicted vorticity $\hat{\omega}_z^{(*)}$ and divergence $\hat{D}^{(*)}$ reside at the corners and the centers, respectively. They are used without averaging in the divergence equation (39) and vorticity equation (38) on the D-grid. The pressures on the two grids, denoted by the subscripts $C$ and $D$, are diagnosed at the cell centers with the buoyancy and vertical velocity, but these two pressures are, in principal, different from each other. Since the pressure is a diagnostic variable, no time stamp is included.

We have tested several variations of the predictor-corrector time-integration algorithm outlined above. There are two main reasons for doing so. First, we do not have access to the details of how the Lin and Rood (1997) scheme is implemented, and second, we want to examine the extent to which the solutions depend on the time-integration schemes. We hope that the results of our analysis will provide some guidance to future users of the CD grid. The construction of the five schemes is summarized in Table 1.

Scheme I is the simplest. Its equations are given by Eqs. (32)–(42), with the assumption that $\hat{P} \equiv \hat{P}_C = \hat{P}_D$. To derive the dispersion equations we first eliminate the variables with $(*)$ in the correction-step equations by using the equations from the predictor step. Then we use

$$\hat{\Phi}^{(n+1)} = e^{-i\nu\tau}\hat{\Phi}^{(n)} , \qquad (43)$$

where $\hat{\Phi}$ represents $\hat{\omega}_z$, $\hat{D}$ and $\hat{B}$, to obtain the discrete dispersion relation for complex frequencies as

$$\mu^2\left[\left(e^{-i\nu\tau} - \sigma_N\right)^2 + \tau^2 N^2\right]L^2 + \left[\left(e^{-i\nu\tau} - \sigma_f\right)^2 + \mu^2\tau^2 f^2\right]\sigma_m^2 = 0 , \qquad (44)$$



where we use the following definitions to shorten the equation:

$$\sigma_N \equiv 1 - \tfrac{1}{2}\tau^2 N^2 \ , \ \sigma_f \equiv 1 - \tfrac{1}{2}\tau^2 f^2 \ \text{and} \ \sigma_m^2 \equiv m^2 + \frac{1}{4H^2} \ . \tag{45}$$

The (complex) dispersion equation (44) is obtained by requiring nontrivial solutions for $\hat{D}$. The nontrivial solutions for $\hat{\omega}_z$ and $\hat{B}$ produce an additional mode, satisfying $e^{-i\nu\tau} - 1 = 0$, which corresponds to a steady and neutral mode associated with

the steady balanced state. By using $e^{-i\nu\tau} - 1 = 0$ in Eq. (44), where $\nu_r$ and $\nu_i$ are the frequency and the rate of amplification/decay of a mode, respectively, we obtain the dispersion relations that govern the frequency and the amplification factor as

$$e^{2\nu_i\tau}\left(\mu^2 L^2 + \sigma_m^2\right)\cos\left(2\nu_r\tau\right) - 2e^{\nu_i\tau}\left(\mu^2 \sigma_N L^2 + \sigma_f \sigma_m^2\right)\cos\left(\nu_r\tau\right)$$

$$+\left(\sigma_N^2 + \tau^2 N^2\right)\mu^2 L^2 + \left(\sigma_f^2 + \tau^2\mu^2 f^2\right)\sigma_m^2 = 0 \ , \tag{46a}$$

and

$$e^{\nu_i\tau} = \frac{2\left(\mu^2\sigma_N L^2 + \sigma_f\sigma_m^2\right)}{\left(\mu^2 L^2 + \sigma_m^2\right)}\frac{\sin\left(\nu_r\tau\right)}{\sin\left(2\nu_r\tau\right)} \ , \tag{46b}$$

respectively.

A Newton-Raphson iteration is used to find the frequency $\nu_r$ that satisfies Eqs. (46a) and (46b) simultaneously. As a

check, we also use a simple search algorithm to find the frequency. The two methods give the virtually the same unique solution unless the time step $\tau$ is very large. Although we obtain real frequencies for moderately unstable amplification factors ($e^{\nu_i\tau} > 1$), we limit the discussion here to near-neutral cases ($e^{\nu_i\tau} \approx 1$). Before discussing the results, we mention that the dispersion relations given by Eqs. (46a) and (46b) yield a solution satisfying $\nu_r = 0$ for the SRHS ($\mu = 0$), which indicates that the discrete solutions include a dynamically inert mode. The solutions of Eq. (46b) with $\nu_r \to 0$ and $\mu = 0$

show that this mode is very weakly damped ($e^{\nu_i\tau} = \sigma_f < 1$). Recall that $\sigma_f \equiv 1 - \tfrac{1}{2}\tau^2 f^2$, so that the damping rate can be considered negligible with the time steps typically used in atmosphere models.





The steady balanced solution can be obtained by assuming nondivergent motion ( $\hat{D}^{(n+1)} = \hat{D}^{(n)} = 0$ ) and a steady and quasi-static state ( $\hat{\omega}_z^{(n+1)} - \hat{\omega}_z^{(n)} = 0$ and $\hat{B}^{(n+1)} - \hat{B}^{(n)} = 0$ ). To satisfy Eq. (42), the vertical velocity must also vanish ( $\hat{w}^{(n+1)} = \hat{w}^{(n)} = 0$ ) in the balanced state. We obtain the unique balanced solution given by

$$f\hat{\omega}_z = -L^2\hat{P}, \quad f\left(m^2 + \frac{1}{4H^2}\right)\hat{\omega}_z = \left(\underset{\sim}{i}m - \frac{1}{2H}\right)L^2\hat{B} \quad \text{and} \quad \left(\underset{\sim}{i}m + \frac{1}{2H}\right)\hat{P} = \hat{B}. \tag{47}$$

These are the discrete equivalents of the continuous relations given by Eqs. (9a)–(9c). We conclude that the balanced-state solution is maintained by Scheme I.

**Results.** The results presented in this section, as inferred from our normal-mode analyses, demonstrate that the dispersion of the inertia-gravity waves produced on the CD-grid with various time-integration schemes is in all cases close to the one produced by the D-grid. This is in agreement with the conclusions of Skamarock (2008). For some of the time-integration

choices (not discussed here), we could not uniquely determine a steady balanced state solution. For some choices, we could not even find real frequencies.

The frequency plots for the inertia-gravity modes on the CD grid with Scheme I are shown in Fig. 5. The time steps $\tau$ used in computing these frequencies are $50\,\text{s}$, $120\,\text{s}$, $150\,\text{s}$ and $210\,\text{s}$ for the 2-km, 10-km, 25-km and 100-km cases, respectively. Schemes II, III, IV and V produce very similar frequency plots (not shown here, but included in the

supplementary material). All of these plots show a strong qualitative resemblance to those for the D grid shown in Fig. 4, and therefore, the discussion presented for the D-grid case can be applied to the CD grid as well, and is omitted here for brevity.

### 3.4.1 Solutions on the DC grid

Our normal-mode analysis shows that, in agreement with the results of Skamarock (2008), the CD grid behaves similarly to the D-grid rather than the C grid. One possible explanation is that the dispersion of the inertia-gravity waves on the CD-grid

is almost entirely dominated by the corrector step. To test this hypothesis, we constructed a "DC-grid" by reversing the integration sequence used in the CD-grid. With the DC-grid, the D and C grids are used in the predictor and corrector steps, respectively. One of the solutions that corresponds to Scheme II (discussed above) produces a C-grid like discrete dispersion relation for the DC-grid sequence. A detailed discussion of the DC-grid can be found in the supplementary material.

We do not advocate the use of the DC-grid because we do not think it has any advantage over the C-grid, and the C-grid

is both simpler and computationally less expensive than the DC-grid.

### 3.5 Solutions on the A grid

The A-grid is a completely unstaggered grid, on which the mass and the zonal and meridional wind components are predicted/diagnosed at the same grid points (see Fig. 1e). The quadrilateral A-grid can be viewed as the superposition of four





non-interacting C-grids or Z-grids (see the supplementary material). We are not aware of any major model that currently uses the quadrilateral A grid, but the icosahedral A-grid is used by NICAM (Satoh et. al., 2008), and the FIM and NIM models developed at NOAA ESRL (Lee and MacDonald, 2009).

**3.5.1 Discrete dispersion of the inertia-gravity waves on an *f*-plane**

We define the discrete vorticity and divergence from the components of the horizontal velocity on the A-grid shown in Fig. 1e as

$$\left(\omega_z\right)_{i,j} \equiv \frac{1}{2d}\left(v_{i+1,j} - v_{i-1,j} - u_{i,j+1} + u_{i,j-1}\right),$$ (48a)

and

$$D_{i,j} \equiv \frac{1}{2d}\left(u_{i+1,j} - u_{i-1,j} + v_{i,j+1} - v_{i,j-1}\right),$$ (48b)

respectively. The vertically and temporally continuous and horizontally discrete versions of the vorticity and divergence equations, given by Eqs. (2) and (3), respectively, can be written as

$$\frac{\partial}{\partial t}\left(\omega_z\right)_{i,j} = -fD_{i,j},$$ (49)

and

$$\frac{\partial}{\partial t}D_{i,j} = f\left(\omega_z\right)_{i,j} - \frac{1}{4d^2}\left(P_{i+2,j} + P_{i-2,j} + P_{i,j+2} + P_{i,j-2} - 4P_{i,j}\right).$$ (50)

By including Eq. (15), which is also the discrete version of Eq. (7) on the A grid, we obtain the discrete dispersion relation for the inertia-gravity modes on the A-grid (following derivations parallel to those for the Z, C and D grids) as

$$v^2 = \frac{N^2\left(\tilde{\xi}^2 k^2 + \tilde{\eta}^2 \ell^2\right) + f^2\left(m^2 + \frac{1}{4H^2}\right)}{\left(\tilde{\xi}^2 k^2 + \tilde{\eta}^2 \ell^2\right) + \left(m^2 + \frac{1}{4H^2}\right)},$$ (51)

where

$$\tilde{\xi} \equiv \frac{\sin(kd)}{kd} \quad \text{and} \quad \tilde{\eta} \equiv \frac{\sin(\ell d)}{\ell d}.$$ (52)





The dispersion relation for the A-grid resembles that of the Z-grid except that the factors $\xi$ and $\eta$ that arise with the Z-grid are replaced by $\tilde{\xi}$ and $\tilde{\eta}$, respectively. The factors $\tilde{\xi}$ and $\tilde{\eta}$ become zero for the SRHS ($kd = \ell d = \pi$), which yields $\nu = f$ for all vertical scales. The modes with the SRHS oscillate as purely inertial modes. In the absence of rotation, these modes do not propagate or oscillate. Although it cannot be detected by the normal-mode analysis, the A grid generates

multiple, linearly decoupled (or nonunique) solutions. This point is discussed further in subsection 3.8. We will show the plots of the frequencies obtained with the A grid in subsection 3.9, along with the corresponding plots for the E and B grids.

### 3.6 Solutions on the E grid

The E-grid can be viewed as the superposition of two C-grids, in which the cell centers of one C-grid are placed at the corners of a second C-grid, as indicated by solid boundary lines in Fig. 1f. On the resulting E-grid, the zonal and meridional

components of the horizontal velocity are predicted at the same points; this can be seen as an advantage over the C-grid. As noted by Arakawa and Lamb (1977), the E-grid is identical to a B-grid rotated by 45° (shown in Fig. 1g) with a grid spacing of $d/\sqrt{2}$. The borders of the cells of the B-grid are indicated by the dashed lines in Fig. 1f. We will see next that from the vorticity and divergence point of view the E grid can also be viewed as a superposition of two independent and noninteracting Z grids.

The E-grid was used in operational models of the U. S. National Centers for Environmental Prediction for more than three decades (Janjic, 1984). We are not aware of any model that currently uses the E grid.

#### 3.6.1 Discrete dispersion of the inertia-gravity waves on an *f*-plane

We first define the discrete vorticity and divergence from the components of the horizontal velocity for the integer ($i, j$) and half-integer ($i+1/2, j+1/2$) center points of the E-grid shown in Fig. 1f. We than write the discrete versions of the

divergence, vorticity and mass equations, respectively Eqs. (2), (3) and (7), for the integer and half-integer points as follows:

Equations for the centers (integer points) of E-grid:    Equations for the centers (half-integer points) of E-grid:

$$\left(\omega_z\right)_{i,j} \equiv \frac{1}{d}\left(v_{i+1/2,j} - u_{i,j+1/2} - v_{i-1/2,j} + u_{i,j-1/2}\right),\quad(53a)$$
$$\left(\omega_z\right)_{i+1/2,j+1/2} \equiv \frac{1}{d}\left(v_{i+1,j+1/2} - u_{i+1/2,j+1} - v_{i,j+1/2} + u_{i+1/2,j}\right),\ (57a)$$

$$D_{i,j} \equiv \frac{1}{d}\left(u_{i+1/2,j} - v_{i,j-1/2} + v_{i,j+1/2} - u_{i-1/2,j}\right),\quad(53b)$$
$$D_{i+1/2,j+1/2} \equiv \frac{1}{d}\left(u_{i+1,j+1/2} - v_{i+1/2,j} + v_{i+1/2,j+1} - u_{i,j+1/2}\right),\quad(57b)$$

$$\frac{\partial}{\partial t}\left(\omega_z\right)_{i,j} = -fD_{i,j},\quad(54)$$
$$\frac{\partial}{\partial t}\left(\omega_z\right)_{i+1/2,j+1/2} = -f D_{i+1/2,j+1/2},\quad(58)$$





$$\frac{\partial D_{i,j}}{\partial t} = f\left(\omega_z\right)_{i,j} - \frac{1}{d^2}\left(P_{i+1,j} + P_{i-1,j} + P_{i,j+1} + P_{i,j-1} - 4P_{i,j}\right), \quad (55)$$

$$\frac{\partial D_{i+1/2,j+1/2}}{\partial t} = f\left(\omega_z\right)_{i+1/2,j+1/2} - \frac{1}{d^2}\left(P_{i+3/2,j+1/2} + P_{i-1/2,j+1/2}\right.$$
$$\left. + P_{i+1/2,j+3/2} + P_{i+1/2,j-1/2} - 4P_{i+1/2,j+1/2}\right), \quad (59)$$

$$\left(\frac{\partial^2}{\partial t^2} + N^2\right)D_{i,j} = -\left(m^2 + \frac{1}{4H^2}\right)\frac{\partial P_{i,j}}{\partial t}, \quad (56)$$

$$\left(\frac{\partial^2}{\partial t^2} + N^2\right)D_{i+1/2,j+1/2} = -\left(m^2 + \frac{1}{4H^2}\right)\frac{\partial P_{i+1/2,j+1/2}}{\partial t}, \quad (60)$$

By using Eq. (16) or Eq. (22) separately in Eqs. (54)–(56) and Eqs. (58)–(60), we obtain the same discrete dispersion relation for the inertia-gravity modes at the integer and half-integer points as

$$v^2 = \frac{N^2\left(\xi^2 k^2 + \eta^2 \ell^2\right) + f^2\left(m^2 + \frac{1}{4H^2}\right)}{\left(\xi^2 k^2 + \eta^2 \ell^2\right) + \left(m^2 + \frac{1}{4H^2}\right)}, \quad (61)$$

where the factors $\xi$ and $\eta$ are defined by Eq. (18). The dispersion equation (61) is identical to that of the Z-grid given by

Eq. (17). The important difference with the E-grid, however, is that there are two solutions, one for the integer points and the

other for the half-integer points. This can be seen by a comparison of the discrete equations for the integer points Eqs. (54)–(56) with the corresponding equations for the half-integer points Eqs. (58)–(60). The discrete equations for the integer points use only information from integer points, and those for the half integer points use only information from the half-integer points. Of course, this means that the solution includes computational modes arising from multiple (or nonunique) solutions. Further discussion of the computational-mode on the E-grid is given in subsection 3.8.

**3.7 Solutions on the B grid**

On the B-grid, the mass and thermodynamic variables are predicted at the cell centers while the horizontal velocity components are predicted at the cell corners (see Fig. 1g). The fact that the two components of the horizontal velocity are predicted at the same points is convenient for the Coriolis terms of the momentum equations. In the linearized system considered here, the vorticity and divergence are both predicted at the cell centers.

A version of the Goddard Institute for Space Studies global circulation model used the B grid (Hansen et al., 1983). The B grid has been used in many ocean models and can perform as well as the C grid in ocean models because the Rossby radius of deformation can be smaller than the grid spacing in ocean applications (Randall, 1994 and Arakawa and Lamb, 1977). This is now changing as ocean models move to higher horizontal resolutions (e.g., Adcroft et al., 2016).



### 3.7.1 Discrete dispersion of the inertia-gravity waves on an *f*-plane

We define the discrete versions of the vorticity and divergence on the B-grid shown in Fig. 1g as

$$(\omega_z)_{i,j} \equiv \frac{1}{2d}\left( \upsilon_{i+1/2,j+1/2} + \upsilon_{i+1/2,j-1/2} - \upsilon_{i-1/2,j+1/2} - \upsilon_{i-1/2,j-1/2} \right)$$

$$-\frac{1}{2d}\left( u_{i+1/2,j+1/2} + u_{i-1/2,j+1/2} - u_{i+1/2,j-1/2} - u_{i-1/2,j-1/2} \right), \tag{62a}$$

5   and

$$D_{i,j} \equiv \frac{1}{2d}\left( u_{i+1/2,j+1/2} + u_{i+1/2,j-1/2} - u_{i-1/2,j+1/2} - u_{i-1/2,j-1/2} \right)$$

$$+\frac{1}{2d}\left( \upsilon_{i+1/2,j+1/2} + \upsilon_{i-1/2,j+1/2} - \upsilon_{i+1/2,j-1/2} - \upsilon_{i-1/2,j-1/2} \right), \tag{62b}$$

respectively. The vorticity equation is identical to that of the Z-grid Eq. (13) and E-grid Eq. (56). The divergence equation for the B-grid is

$$\frac{\partial}{\partial t} D_{i,j} = f\left(\omega_z\right)_{i,j} - \frac{1}{2d^2}\left( P_{i+1,j+1} + P_{i+1,j-1} + P_{i-1,j+1} + P_{i-1,j-1} - 4P_{i,j} \right). \tag{63}$$

The discrete Laplacian in the divergence equation (63) uses information from the cells sharing corners instead of walls, which distinguishes the B-grid from of the all other grids considered here. The B-grid also uses the same mass equation as the Z-grid Eq. (15) and E-grid Eq. (56), which are also used with C and A grids. By using Eq. (16) in the vorticity Eq. (13), divergence Eq. (63) and mass equation (56), we obtain the discrete dispersion relation for the inertial-gravity modes of the

15   B-grid as

$$v^2 = \frac{N^2\left( \xi^2 k^2 + \eta^2 \ell^2 - \frac{1}{2}d^2\xi^2 k^2\eta^2\ell^2 \right) + f^2\left( m^2 + \frac{1}{4H^2} \right)}{\left( \xi^2 k^2 + \eta^2 \ell^2 - \frac{1}{2}d^2\xi^2 k^2\eta^2\ell^2 \right) + \left( m^2 + \frac{1}{4H^2} \right)}, \tag{64}$$

where the factors $\xi$ and $\eta$ are defined by Eq. (18). The Laplacian term $\xi^2 k^2 + \eta^2 \ell^2 - \frac{1}{2}d^2\xi^2 k^2\eta^2\ell^2$ becomes zero for the smallest resolvable horizontal scales $\left( kd = \ell d = \pi \right)$, which leads to $v = f$ for all vertical scales. As with the A-grid, the modes with the smallest resolvable horizontal scales appear as purely inertial modes. There are also multiple solutions with

20   the B grid, as well as computational modes, which will be discussed next.





### 3.8 Computational modes on the A, B and E grids

The most easily recognizable feature of the computational modes is the dynamically inert modes for the SRHS. Consider the Laplacian operators of the A and B grids used on the right hand sides of Eqs. (50) and (63), respectively. The Laplacian operator of the A grid yields a zero result for patterns (1), (2), (3) and (4) shown in Fig. 6, presuming that the red and blue
colors correspond to same magnitudes with opposite signs (also see Fig. 1e). These patterns are caused by the existence of four independent solutions as demonstrated in the supplementary material. Because these four patterns give a Laplacian of zero, they remain unchanged. They are the computational modes of the A grid. On the other hand, the Laplacian of the B grid gives zero results for pattern (3), and therefore, this pattern is the computational mode of the B grid. The solutions on the red and blue grid networks are completely decoupled from each other on the A and B grids (also see Fig. 1g). For a
rotating case, the A and B grids produce inertial oscillations at all grid points.

     The E grid holds two decoupled solutions, one for the network of centers (say red cells) governed by Eqs. (52)–(56) and the other for the network of corners (say blue cells) governed by Eqs. (58)–(60). Pattern (5) is the computational mode of the E grid. These decoupled solutions are also indicated by different colors in Fig. 1f.

     The A grid can produce four independent and noninteracting (non-unique) solutions as indicated with different colors in
Fig. 1e, and the B and E grids can produce two independent and noninteracting (non-unique) solutions as indicated with different colors in Figs. 1f and g, respectively. To avoid the separation of solutions, a horizontal mixing process such as diffusion is needed, but such a process can have adverse side effects such as unrealistic dissipation of the small-scale physical modes.

     The Z, C, D and CD grids do not have independent and noninteracting solutions like those of the A, B and E grids. The
C, D and CD grids do have dynamically inert modes for the SRHS because of the averaging of the variables.

### 3.9 An illustrative discussion of the dispersion of modes obtained with the A, E and B grids

The dispersion relations of the A-, E- and B-grid solutions are free from the averaging factor $\mu$ (see Eqs. (51), (61) and (64)) because these grids do not require averaging in their divergence and mass equations. All these grids yield $v = f$ at their SRHS, and consequently their frequency plots resemble each other. The frequency plots of the inertia-gravity modes for the
A, E and B grids are shown in Figs. 1e, f, and g, respectively. The horizontal grid spacing $\sqrt{2}d$ is used in the E-grid plots to maintain the same density of cell centers as the other grids. The frequency of all vertical scales, particularly deep vertical modes, makes a very sharp change near the SRHS. This sharp change causes the group velocity to reverse severely near the SRHS for all vertical modes.

     In light of the discussion in the previous subsection, we conclude that the frequencies of the modes obtained using the A,
E and B grids are not unique. Solutions as shown in Fig. 6 at the neighboring grid points can be decoupled from each other



although they satisfy the same dispersion relation.

The conclusions of our normal-mode analysis of the nonhydrostatic anelastic inertia-gravity wave solutions obtained with the A, E and B grids are not substantially different from the conclusions of earlier studies of the normal-modes of the shallow-water equations as described by many authors, including Arakawa and Lamb (1977) and Randall (1994).

**4. Vertical discretization of linear anelastic equations on the L- and CP-grids and discrete dispersion equations**

There are two vertical grids available for the vertical discretization of models based on the height and pressure vertical coordinates. These are the Lorenz grid (or L-grid; Lorenz, 1960) and the Charney-Phillips (or CP-grid; Charney-Phillip, 1953). The vertical grid used with isentropic coordinates is analogous to the CP grid (see Konor and Arakawa, 1997 and 2000; Thuburn and Woolings et al., 2005; and Toy and Randall, 2007). The difference between the L and CP grids is that the

L-grid predicts the thermodynamic variables (in this paper the buoyancy) at the model "layers", which are denoted by the dashed lines in Fig. 10, along with the horizontal momentum or vorticity and divergence and mass variable or pressure. The CP-grid, on the other hand, predicts the thermodynamic variables at the model "interfaces", which are the solid lines in Fig. 10, along with the vertical momentum.

The Lorenz (1960) grid is the most commonly used vertical grid for atmospheric models. Notable examples of the

nonhydrostatic models that use the L-grid are MPAS (Skamarock et al., 2012), and NICAM (Satoh et al., 2008). Many studies (e.g. Arakawa and Moorthi, 1988; Hollingsworth, 1995; Arakawa and Konor, 1996) have raised issues with the L-grid. All of these studies find evidence of a dynamically inert mode in the vertical structure of the potential temperature. Girard et al. (2014) report an unusually high number of occurrence of the 2-dz wave in the vertical structure of simulations with GEM (a nonhydrostatic model), which is consistent with a dynamically inert mode. Arakawa and Moorthi (1988) and

Arakawa and Konor (1996) additionally find evidence of a spurious rapid growth of short baroclinic modes in the L-grid models. We will not discuss this subject further because it is outside the scope of this paper.

The alternative to the L-grid is the CP-grid, which is free of these problems. Examples of nonhydrostatic models that use the CP-grid are the UK Met Office models (Wood et al., 2014) and GEM (Girard et al., 2014). Toy and Randall (2009) also use the CP-grid in their vertical discretization.

Our purpose in this section is to present a comparison of the L and CP grids with the same normal-mode analysis that we used for the comparison of the horizontal grids, so that we can assess the relative impact of the horizontal and vertical grids on the solutions of the nonhydrostatic anelastic equations.





### 4.1 L grid

The vertically discrete, temporally and horizontally continuous versions of the vorticity Eq. (2), divergence Eq. (3), vertical momentum Eq. (4), thermodynamic Eq. (5) and continuity Eq. (6) equations on the L-grid shown in Fig. 10 can be written as

$$\frac{\partial(\omega_z)_k}{\partial t} = -f D_k, \tag{65}$$

$$\frac{\partial D_k}{\partial t} = f(\omega_z)_k - \nabla_{\mathrm{H}}^2 P_k, \tag{66}$$

$$\frac{\partial w_{k+1/2}}{\partial t} = -\frac{P_{k+1}-P_k}{\delta z} + \left(\frac{1}{2\rho_0}\frac{\partial \rho_0}{\partial z}\right)\frac{P_{k+1}+P_k}{2} + \frac{B_k + B_{k+1}}{2}, \tag{67}$$

$$\frac{\partial B_k}{\partial t} = -N^2 \frac{w_{k+1/2}+w_{k-1/2}}{2}, \tag{68}$$

and

$$D_k + \frac{w_{k+1/2}-w_{k-1/2}}{\delta z} + \left(\frac{1}{2\rho_0}\frac{\partial \rho_0}{\partial z}\right)\frac{w_{k+1/2}+w_{k-1/2}}{2} = 0, \tag{69}$$

10   respectively. Recall that $(1/\rho_0)(\partial \rho_0/\partial z) = -1/H$ for an isothermal atmosphere. By using

$$\phi_k(x,y,t) = \mathrm{Re}\left\{\hat{\Phi}e^{i(kx+\ell y+m\delta z k - vt)}\right\} \quad \text{and} \quad \phi_{k+1/2}(x,y,t) = \mathrm{Re}\left\{\hat{\Phi}e^{i[kx+\ell y+m\delta z(k+1/2)-vt]}\right\} \tag{70}$$

in Eqs. (65)–(69), we obtain the discrete dispersion relation for the inertia-gravity modes on the L-grid as

$$v^2 = \frac{\mu_z^2 N^2 \left(k^2+\ell^2\right) + f^2 \left(\zeta^2 m^2 + \mu_z^2 \frac{1}{4H^2}\right)}{\left(k^2+\ell^2\right) + \left(\zeta^2 m^2 + \mu_z^2 \frac{1}{4H^2}\right)}, \tag{71}$$

where

$$\zeta \equiv \frac{1}{\frac{1}{2}m\delta z}\sin\left(\tfrac{1}{2}m\delta z\right) \quad \text{and} \quad \mu_z \equiv \cos\left(\tfrac{1}{2}m\delta z\right). \tag{72}$$

The vertical averages of buoyancy in the vertical momentum equation (67) and the vertical velocity in the thermodynamic equation (68) both result in the appearance of the averaging factor $\mu_z^2$ in the gravity term, which prevents





the smallest vertical scale ( $m\delta z = \pi$ ) from recognizing the stability because $\mu_{\hat{z}}^2 = 0$ for this scale. This permits a dynamically inert solution in which the buoyancy is completely decoupled from the rest of the variables. The frequency obtained from Eq. (71) in this case applies to the variables other than the buoyancy. The term $1/4H^2$ is multiplied by the averaging factor $\mu_{\hat{z}}^2$ because the pressure and vertical velocity are vertically averaged in Eqs. (67) and (69), respectively,

5    which slightly modifies the dispersion.

**4.2 CP grid**

The vertically discrete, and temporally and horizontally continuous versions of the vorticity, divergence and continuity equations on the CP-grid shown in Fig.10 are given by Eqs. (65), (66) and (69), and the vertical momentum and thermodynamic equations can be written as

$$\frac{\partial w_{k+1/2}}{\partial t} = -\frac{P_{k+1} - P_k}{\delta z} + \left( \frac{1}{2\rho_0} \frac{\partial \rho_0}{\partial z} \right) \frac{P_{k+1} + P_k}{2} + B_{k+1/2} ,$$ (73)

and

$$\frac{\partial B_{k+1/2}}{\partial t} = -N^2 w_{k+1/2} ,$$ (74)

respectively. Following the procedure for the L-grid case, we obtain the discrete dispersion relation for the inertia-gravity modes as

$$\nu^2 = \frac{N^2 \left( k^2 + \ell^2 \right) + f^2 \left( \zeta^2 m^2 + \mu_{\hat{z}}^2 \frac{1}{4H^2} \right)}{\left( k^2 + \ell^2 \right) + \left( \zeta^2 m^2 + \mu_{\hat{z}}^2 \frac{1}{4H^2} \right)} ,$$ (75)

where $\mu_{\hat{z}}^2$ and $\zeta^2$ are given by Eq. (72). The main difference between the dispersion equation for the CP-grid and that for the L-grid is that the averaging factor $\mu_{\hat{z}}^2$ does not appear in the gravity term of the CP-grid dispersion equation given by Eq. (75), and thus the dynamically inert mode obtained with the L-grid does not exist on the CP-grid.

Figure 11 shows the frequencies as functions of composite horizontal wavenumber of inertia-gravity modes obtained with
20    the L- and CP-grids. The true frequencies are also shown in separate panels of the figure. The figure shows the results for two vertical integer wavenumbers (or number of layers), namely $n_{max} = 320$ and $80$. We included additional frequency lines corresponding to more integer vertical wavenumbers than were used in the plots of Sect. 3 (indicated by thinner solid lines in





the plots). In the L-grid solutions shown in Fig. 11. b and e, the frequency of the smallest vertical resolvable mode, identified by $n_{max}$, deviates greatly from the true frequency, which yields values equal to or less than inertial frequency. As the vertical scale approaches the smallest resolvable scale, the modes gradually lose their ability to recognize the effects of buoyancy. For the mode with the smallest scale, the buoyancy is completely decoupled from the wind field; for that mode, the buoyancy is dynamically inert. The group velocity of the mode is reversed for the short horizontal scales. In contrast, the frequency of the CP-grid solutions is generally close to the true frequency, but slightly higher.

Because the frequency errors with the L-grid increase with increasing horizontal wavenumber, we can expect that the adverse effect of the use of the L-grid on the dispersion of the inertia-gravity waves will be more problematic in high-resolution nonhydrostatic applications than in low-resolution quasi-hydrostatic ones.

**5. A summary of the discrete dispersion equations**

Table 2 summarizes the discrete dispersion relations obtained for the horizontal and the vertical grids. The range of the horizontal and vertical wavenumbers normalized by the horizontal and vertical grid spacing is indicated for each dispersion equation.

**6. Numerical solutions**

In this section, we first discuss the construction of linear anelastic numerical models based on the various horizontal grids, and then present analyses of simulations of inertia-gravity modes on a middle-latitude $f$-plane. Our purpose is to confirm the results of the normal-mode analyses of the discrete equations, and to investigate issues that are not completely revealed by the normal-mode analysis. In the main text, we discuss only results for the Z, C, D, and CD grids. Additional discussion for the A, B, and E grids is given in the supplementary material.

**6.1 Equations of the C, D and CD grid models**

We write the horizontally discrete version equations Eqs. (2)–(6) and Eq. (8) on the C and D grids as

Equations of the C-grid model:

$$\frac{\partial\left(\omega_z\right)_{i+1/2,j+1/2}}{\partial t} = -f\overline{D}_{i+1/2,j+1/2} , \qquad (76)$$

$$\frac{\partial D_{i,j}}{\partial t} = f\overline{\left(\omega_z\right)}_{i,j} - \left(\tilde{\nabla}_H^2 P\right)_{i,j} , \qquad (77)$$

Equations of the D-grid model:

$$\frac{\partial\left(\omega_z\right)_{i,j}}{\partial t} = -f\overline{D}_{i,j} , \qquad (81)$$

$$\frac{\partial D_{i+1/2,j+1/2}}{\partial t} = f\overline{\left(\omega_z\right)}_{i+1/2,j+1/2} - \left(\tilde{\nabla}_H^2 P\right)_{i+1/2,j+1/2} , \qquad (82)$$





$$\frac{\partial \tilde{B}_{i,j}}{\partial t} = N^2 D_{i,j} \, , \qquad (78)$$

$$\left(\tilde{\nabla}_{\mathrm{H}}^2 P\right)_{i,j} - \left(m^2 + \frac{1}{4H^2}\right)P_{i,j} = f\overline{\left(\omega_z\right)}_{i,j} + \tilde{B}_{i,j} \, , \qquad (79)$$

where

$$\left(\tilde{\nabla}_{\mathrm{H}}^2 P\right)_{i,j} \equiv \frac{1}{d^2}\left(P_{i+1,j} + P_{i-1,j} + P_{i,j+1} + P_{i,j-1} - 4P_{i,j}\right), \qquad (80)$$

$$\tilde{B}_{i,j} \equiv \left(\frac{\partial}{\partial z} - \frac{1}{2H}\right)B_{i,j} \, ,$$

$$\overline{D}_{i+1/2,j+1/2} \equiv \frac{1}{4}\left(D_{i,j} + D_{i+1,j} + D_{i,j+1} + D_{i+1,j+1}\right),$$

and

$$\overline{\left(\omega_z\right)}_{i,j} \equiv \frac{1}{4}\Big[\left(\omega_z\right)_{i-1/2,j-1/2} + \left(\omega_z\right)_{i+1/2,j-1/2} \\ + \left(\omega_z\right)_{i-1/2,j+1/2} + \left(\omega_z\right)_{i+1/2,j+1/2}\Big].$$

$$\frac{\partial \tilde{B}_{i,j}}{\partial t} = N^2 D_{i,j} \, , \qquad (83)$$

$$\left(\tilde{\nabla}_{\mathrm{H}}^2 P\right)_{i+1/2,j+1/2} - \left(m^2 + \frac{1}{4H^2}\right)P_{i+1/2,j+1/2} \\ = f\overline{\left(\omega_z\right)}_{i+1/2,j+1/2} + \tilde{B}_{i+1/2,j+1/2} \, , \qquad (84)$$

where

$$\left(\tilde{\nabla}_{\mathrm{H}}^2 P\right)_{i+1/2,j+1/2} \equiv \frac{1}{d^2}\Big(P_{i+3/2,j+1/2} + P_{i-1/2,j+1/2} + \\ + P_{i+1/2,j+3/2} + P_{i+1/2,j-1/2} - 4P_{i+1/2,j+1/2}\Big), \qquad (85)$$

$$\tilde{B}_{i,j} \equiv \left(\frac{\partial}{\partial z} - \frac{1}{2H}\right)B_{i,j} \, ,$$

$$\overline{D}_{i,j} \equiv \frac{1}{4}\left(D_{i+1/2,j+1/2} + D_{i-1/2,j+1/2} + D_{i+1/2,j-1/2} + D_{i-1/2,j-1/2}\right),$$

and

$$\overline{\left(\omega_z\right)}_{i+1/2,j+1/2} \equiv \frac{1}{4}\Big[\left(\omega_z\right)_{i+1,j+1} + \left(\omega_z\right)_{i,j+1} + \left(\omega_z\right)_{i+1,j} + \left(\omega_z\right)_{i,j}\Big].$$

The C-grid involves averaging twice. First we must average $D$ from the centers to the $\omega_z$-points at the cell corners for use in the vorticity equation, Eq. (76), and second we must average $\omega_z$ from corners to the $D$-points at the cell centers for use

in the divergence equation, Eq. (77).

Consider the perturbation patterns consisting of vertical and horizontal stripes, or a checkerboard for the buoyancy $\tilde{B}$ (or divergence $D$) on the C-grid. Such a perturbation given to $D$ at the cell centers cannot be recognized at the cell corners where $\omega_z$ is predicted. These patterns therefore behave as pure gravity modes rather than inertia-gravity modes. We interpret these as physical modes behaving badly.

The D-grid requires four separate four-point averagings in four equations. Two of these are the averaging of $D$ from corners to the $\omega_z$- and $B$-points at cell centers. The third one is the averaging of $\omega_z$ from the centers to the $D$-points at the corners. The fourth one is the averaging of $B$ to the corners, for use on the right-hand side of the elliptic equation. In comparison to the C-grid, the D-grid requires two additional averagings in the continuity equation and in the elliptic equation





that determines the pressure, $P$. We can avoid averaging in the continuity equation, but then we have to use averaging in the thermodynamic equation. If the perturbation patterns mentioned above are given on the D grid, they cannot be recognized, and so they appear as stationary, dynamically inert patterns.

Now we write the temporally and horizontally discrete version equations Eqs. (2)–(6) and Eq. (8) on the CD grids as

5    Predictor step (*) on the C-grid:

$$\left(\omega_z\right)^{(*)}_{i+1/2,j+1/2} = \overline{\left(\omega_z\right)}^{(n)}_{i+1/2,j+1/2} - \tfrac{1}{2}\tau f D^{(n)}_{i+1/2,j+1/2}, \tag{86}$$

$$D^{(*)}_{i,j} = \overline{D}^{(n)}_{i,j} + \tfrac{1}{2}\tau\left[\underline{f\overline{\left(\omega_z\right)}^{(*)}_{i,j}} - \left(\tilde{\nabla}^2_{\mathrm{H}}P\right)_{i,j}\right], \tag{87}$$

$$\tilde{B}^{(*)}_{i,j} = \tilde{B}^{(n)}_{i,j} + \tfrac{1}{2}\tau N^2 \overline{D}^{(n)}_{i,j}, \tag{88}$$

$$\left(\tilde{\nabla}^2_{\mathrm{H}}P\right)_{i,j} - \left(m^2 + \frac{1}{4H^2}\right)P_{i,j} = \underline{f\overline{\left(\omega_z\right)}^{(*)}_{i,j}} + \tilde{B}^{(*)}_{i,j}, \tag{89}$$

Corrector step (n+1) on the D-grid:

$$\left(\omega_z\right)^{(n+1)}_{i,j} = \left(\omega_z\right)^{(n)}_{i,j} - \tau f D^{(*)}_{i,j}, \tag{90}$$

$$D^{(n+1)}_{i+1/2,j+1/2} = D^{(n)}_{i+1/2,j+1/2} + \tau\left[\underline{f\overline{\left(\omega_z\right)}^{(n+1)}} - \left(\tilde{\nabla}^2_{\mathrm{H}}P\right)\right]_{i+1/2,j+1/2}, \tag{91}$$

$$\tilde{B}^{(n+1)}_{i,j} = \tilde{B}^{(n)}_{i,j} + \tau N^2 \underline{\underline{\overline{D}^{(n+1)}_{i,j}}}, \tag{92}$$

$$\left(\tilde{\nabla}^2_{\mathrm{H}}P\right)_{i+1/2,j+1/2} - \left(m^2 + \frac{1}{4H^2}\right)P_{i+1/2,j+1/2} =$$

$$f\underline{\overline{\left(\omega_z\right)}^{(n+1)}_{i+1/2,j+1/2}} + \overline{\tilde{B}}^{(n+1)}_{i+1/2,j+1/2}, \tag{93}$$

where $\left(\tilde{\nabla}^2_{\mathrm{H}}P\right)_{i,j}$ is given by Eq. (80).

In the above equations,

$$\overline{\left(\omega_z\right)}^{(*)}_{i,j} \equiv \frac{1}{4}\Big[\left(\omega_z\right)_{i-1/2,j-1/2} + \left(\omega_z\right)_{i+1/2,j-1/2}$$

where $\left(\tilde{\nabla}^2_{\mathrm{H}}P\right)_{i+1/2,j+1/2}$ is given by Eq. (85).

In the above equations,

$$\overline{\left(\omega_z\right)}^{(n+1)}_{i+1/2,j+1/2} \equiv \frac{1}{4}\Big[\left(\omega_z\right)_{i,j} + \left(\omega_z\right)_{i+1,j} + \left(\omega_z\right)_{i,j+1} + \left(\omega_z\right)_{i+1,j+1}\Big]^{(n+1)},$$

$$+ \left(\omega_z\right)_{i-1/2,j+1/2} + \left(\omega_z\right)_{i+1/2,j+1/2}\Big]^{(*)},$$

$$\overline{\tilde{B}}^{(n+1)}_{i+1/2,j+1/2} \equiv \frac{1}{4}\left(\tilde{B}_{i,j} + \tilde{B}_{i+1,j} + \tilde{B}_{i,j+1} + \tilde{B}_{i+1,j+1}\right)^{(n+1)}.$$

$$\overline{D}^{(n)}_{i,j} \equiv \frac{1}{4}\left(D_{i-1/2,j-1/2} + D_{i+1/2,j-1/2} + D_{i-1/2,j+1/2} + D_{i-1/2,j-1/2}\right)^{(n)}.$$

On the CD-grid, the principal prognostic variables that are denoted by ( n ) and ( n + 1 ), reside on the D grid. The provisional values of variables that are defined on the C grid are denoted by ( * ). The variables $\overline{\left(\omega_z\right)}^{(n)}_{i+1/2,j+1/2}$ in Eq. (86) and $\overline{D}^{(n)}_{i,j}$ in Eq. (87) are averaged to the grid points where they are used on the C-grid at the beginning of the predictor step. At the correction

20    step on the D-grid, the provisional value of $D^{(*)}$ is directly used in the prediction of $\omega_z$.

The equations given above belong to our primary model We also built a secondary model, in which the single-underlined



terms are replaced by $\overline{\left(\omega_z\right)}_{i,j}^{(*)}$, the double-underlined terms are replaced by $\overline{\left(\omega_z\right)}_{i+1/2,j+1/2}^{(*)}$, and the triple underlined term is

replaced by $\overline{D}_{i,j}^{(*)}$. The primary and secondary models are built by closely following Schemes V and II, respectively. These were discussed in Sect. 3. The two models produce identical numerical solutions, as discussed in the next section.

### 6.2 Checking the dispersion equations through numerical integrations

We have integrated the models described above to check the normal-mode solutions discussed in Sect. 3 and to illustrate the behavior of the computational modes associated with the different grids. For the time discretization of the all models other than the CD-grid model, we use a simple forward time-integration scheme that produces virtually neutral (very weakly unstable) solutions with short time steps in all models. We also tested other time-differencing schemes in a single test case. The trapezoidal scheme produced neutral solutions very similar to those obtained with the forward scheme. A forward-

backward predictor-corrector scheme was excessively dissipative, and a second-order Adams-Bashforth scheme was slightly dissipative.

We first present results from the standing oscillation simulations, to look for the frequencies obtained through the normal-mode analyses. The standing oscillations appear stationary because they are produced by the waves that propagate with the same phase speed in all directions on the horizontal plane. In these simulations, the vertical structure is continuous, and the

vertical wavenumber $n$ is prescribed. The simulations start from the initial buoyancy fields shown in Fig. 12, for which we have selected the horizontal wavelengths of $L = 4$ km, $L = 20$ km, $L = 50$ km and $L = 200$ km in both the $x$- and $y$-directions. These wavelengths correspond to the SRHS of $d$=2 km, $d$=10 km, $d$=25 km, and $d$=100 km, respectively, which were examined in connection with the normal-mode analyses. Here we discuss results for $L = 4$ km and $L = 200$ km. Similar discussions for $L = 20$ km and $L = 50$ km can be found in the supplementary material. The vertical wave numbers

used in the simulations are $n$=320, 640 and 1280 for $L = 4$ km; and $n$=80, 160 and 320 for $L = 200$ km. The sensitivity of the numerical solutions to the grid spacing is examined by using three different grid spacings, which are $d = L/80$, $d = L/4$ and $d = L/2$. The initial perturbation field for $\tilde{B}$ (or any other variable) is shown in Fig. 12, for these three horizontal resolutions. With the highest horizontal resolution ($d = L/80$) shown in Fig. 12a, the initial waves are well resolved, and appear smooth. For the grid spacing $d = L/4$, the initial field is not completely resolved. The initial field is

only recognized as ones, zeros and negative ones, which appears like alternating upside and inverted "pyramids" in Fig. 12b. The grid points marked by plus signs ( + ) are at the positive and negative extremes, and at the wall centers and the corners of the (upside and inverted) pyramids. For $d = L/2$, which corresponds to the shortest horizontal grid spacing to resolve the initial perturbation, the pyramid like structure is again visible with the grid points at the extremes of the (upside and inverted) pyramids (Fig. 12c).



The numerical frequencies of the standing oscillations for the high horizontal resolution case ( $d = L/80$ ) simulated by Z-, C-, CD- and D-grid models are given in Table 3. For comparison, we include in the table the true frequency obtained from Eq. (11). The numerical frequencies obtained with the C-, CD-, D-grid models are very close to the true frequencies. This is expected because of the use of high horizontal resolutions in the simulations. All variables oscillate with the same frequency, and there is a unique solution. The numerical frequencies obtained with the CD-grid are identical to those obtained with the D-grid.

We have repeated the same simulation using $d = L/4$ , which is half of the shortest spacing needed to resolve the initial perturbation. The numerical frequencies obtained with the Z-, C-, CD- and D-grid models for the same horizontal resolution are tabulated in Table 4, which shows that all variables at all the grid points oscillate with the same frequency in the Z-grid simulation. In the C-, CD- and D-grid solutions, all variables are also oscillating with the same frequency, but some of the modes have frequencies lower than the inertial frequency (underlined numbers), which indicates that these modes cannot properly recognize the rotation. This is generally due to the averaging of the divergence and vorticity to each other's grid points. In the $L = 4$ km case, the C and Z grids produce similar frequencies, but the CD and D grids produce much lower frequencies. In the $L = 200$ km case, the C grid produces lower frequencies than the Z grid, sometimes even lower than the inertial frequencies. The CD and D grids produce lower frequencies than the C grid. Note that the CD and D grid frequencies are almost identical to each other.

Finally we tabulate results obtained by using the shortest possible horizontal grid spacing ( $d = L/2$ ) to resolve the initial perturbation shown Fig. 12c in Table 5. This is the horizontal grid spacing for which the errors due to finite-differencing are the largest, and the computational modes impact the solutions most strongly. The D- and CD-grid solutions do not oscillate (indicated by zeros) but the Z grid produces oscillations. Averaging of the initial buoyancy perturbation to the divergence points completely wipes out the wave on the D and CD grids. The checkerboard pattern is dynamically inert. Non-oscillating solutions are also obtained by starting from the vorticity perturbations (as indicated by zeros with in the parentheses in Table 5). In the C-grid solution, the buoyancy and divergence recognize the initial perturbation and do produce oscillations, although the vorticity is decoupled from the divergence and buoyancy, due to the averaging of the divergence to the vorticity points. For the short horizontal scales, the frequencies of the buoyancy and divergence in the C-grid solutions are very close to those of the Z-grid solutions. For the long horizontal and short vertical scales, the frequency with which the buoyancy and divergence oscillate in the C-grid solutions is considerably smaller than that of the Z-grid solutions and the inertial frequency (indicated by underlined numbers in Table 5). If the initial perturbation is given to the vorticity instead of the buoyancy in the C, CD and D grid simulations, the solutions are non-oscillatory for all variables (indicated zeros with in the parentheses in Table 5). The Z-grid solution produces frequencies that are close to the true frequencies even though the initial perturbation is poorly resolved.



### 6.3 A numerical simulation of wave propagation

We have also made simulations to demonstrate the behavior of the computational modes during the propagation of inertia-gravity modes with the seven grids under discussion. These simulations start from a positive bell-shaped buoyancy perturbation placed in the middle of the horizontal domain with rapidly decaying amplitude away from the center. We

modified the bell-shaped perturbation by setting it to zero at every other grid point. We refer to this modification as "initial grid-scale noise." The horizontal domain is 280 by 280 grid points, and the horizontal grid spacing is $d = 50$ m , except that for the E-grid we use $d = 70.71$ m . The radius of the perturbation, that is the distance from the peak of the perturbation to where the perturbation becomes zero, is 950 m. The perturbation is continuous in the vertical, with the integer vertical wavenumber $n$=320. Fig. 13 shows the buoyancy field in the 124-by-124 wide corner-end portion of the horizontal domain,

after 100 simulated minutes of integration. For reference, we show the Z-grid result obtained without the superimposed initial grid-scale noise. No major difference can be seen between the Z-grid solutions started with the computational mode and without it in the portion of the domain shown in Fig 13. In the solution with the grid-scale noise, there is, however, a remnant of the initial noise, which takes the form of grid-scale standing inertia-gravity oscillation at and near the center of the domain where the initial perturbation is strongest. This is not visible in Fig. 13 because it is outside of the plotted

domain. The noise gradually subsides in time as the bulk of the initial perturbation propagates outward from the center of the domain. The C-grid simulation result is very close to the Z-grid result. It is evident that the initial noise does not have a permanent effect on the solutions with the Z- and C-grids.

The CD- and D-grid results resemble the C-grid result except that the noise is apparent near the center of the domain in both simulations (see Fig. 13). Since the noise with the SRHS cannot propagate or oscillate, it is a permanent feature of the

CD- and D-grid simulations. By setting the initial perturbation at every other grid point to zero, the dynamically inert computational mode described by pattern (3) in Fig. 6 is put in place on the A-, B- and E-grid. The other solution in these grids recognizes the initial Gaussian perturbation and yields a propagating wave that has similar features to those produces by the Z, C, CD and D grids.

### 6.4 Numerical simulations of waves that are forced by noisy heating patterns

To assess the performance of the Z, C, D and CD grids in the presence of noisy heating patterns, we performed numerical integrations with the linear models discussed in subsection 6.1, by adding a buoyancy forcing (or heating) term $\left( \partial \tilde{B} / \partial t \right)_{forcing}$ to their buoyancy prediction equations. The forcing term is only added to the buoyancy equation at the correction step in the CD grid model. The noisy forcing pattern, which is created using a random number generator and remains unchanged during the integrations, is confined to a circular region at the center of the domain, and it does not produce a net horizontally

averaged heating. The domain and basic state characteristics of the "forced" models are identical to those described earlier in subsection 6.2. The horizontal grid spacing and the vertical wavenumber are 50 m and $n$=320, respectively.  We use



$\left(\partial \tilde{B}/\partial t\right)_{forcing} = \mp 1 \text{ s}^{-3}$ and use no initial perturbations in these simulations. The details are described in the supplementary material.

In Fig. 14, we show the time evolutions of the domain maxima and minima of the buoyancy for the Z, C, D and CD grid simulations. In the Z- and C-grid simulations, the amplitude of buoyancy perturbation does not increase, which implies that

the divergence (or vertical velocity) responds efficiently to counter the forcing (heating). In the D- and CD-grid simulations, the divergence response is not strong enough to prevent an increase of the amplitudes of the buoyancy perturbations. The D- and CD-grid models may need strong explicit diffusion to prevent an excessive accumulation of noise near the source of the noise.

In summary, only the Z grid generates solutions that correspond to those of the continuous system, in which all variables

interact with each other on all scales. The frequencies simulated with the Z grid are remarkably close to the true frequencies.

The numerical simulations obtained with the C grid are overall very close to those with the Z grid. However, the C grid generates solutions, in which vorticity is decoupled form the divergence for the modes with the SRHS.

The D and CD grid results are virtually identical to each other. All the variables, i.e. vorticity, divergence, mass and vertical velocity, for the modes with the SRHS are completely decoupled from each other on the D and CD grids.

The A, B and E grids have multiple solutions for all resolutions, and they produce dynamically inert solutions, in which the variables are decoupled from each other, for the SRHS (discussed in the supplementary material).

**7. The impact of the nonhydrostatic effects on the horizontal grid selection**

Discretization errors have most often been studied using the shallow-water equations. This is justifiable for an assessment of the errors with the Lamb wave and quasi-hydrostatic modes. The dispersion of the inertia-gravity modes for the shallow-

water system is analogous to that for the Lamb wave, except that the fluid depth is multiplied by $\gamma \equiv 1/(1-\kappa)$ in the Lamb wave solution. The shallow-water system can also be used to assess the discretization errors of the quasi-hydrostatic system because in that system the square of the frequency of the inertia-gravity is a linear function of $\left(k^2 + \ell^2\right)$, as it is with the Lamb wave and shallow-water systems. A detailed discussion of the continuous and discrete dispersion equations can be found in the supplementary material.

In this respect, the analogy between the shallow-water (also quasi-hydrostatic) and anelastic systems is weak. The nonhydrostatic effects of the anelastic system, as with the fully compressible, unified and semi-hydrostatic systems, limit the frequency of the inertia-gravity modes to the Brunt-Vaisala frequency, $N$, as can be seen from the red curves in Fig. 15. In contrast, the frequency of the quasi-hydrostatic modes (black lines) and the Lamb wave (dashed green lines) increases





without bound with increasing horizontal wavenumber. It appears that the deeper modes are "more nonhydrostatic" than the shallower ones, in terms of the separation of the anelastic and quasi-hydrostatic frequencies in Fig. 15.

The Z and C grids actually perform better with the nonhydrostatic systems than they do with the quasi-hydrostatic or shallow-water systems. Consider the discrete dispersion equations (17) and (23) for the Z and C grids, respectively. The

finite-difference errors are represented by $\xi^2$ and $\eta^2$, which are close to unity for small wavenumbers and approximately equal to 0.4 at the SHRS. This also means that $\xi^2 k^2 + \eta^2 \ell^2$ is approximately $0.9 \times k^*$ for $\xi \equiv \eta$ at the SHRS. For deep anelastic modes with large composite horizontal wavenumber ($k^*$), the dependence of the frequency on the horizontal wavenumber is negligible, and therefore the error in the frequency due to finite-differencing is small for both the Z and C grids. As mentioned earlier, the averaging of the Coriolis term with the C grid does not significantly influence the solution

for sufficiently high horizontal wavenumbers.

In conclusion, the Z and C grids perform better with the nonhydrostatic equations than the quasi-hydrostatic or shallow-water equations, particularly in terms of their ability to resolve deep modes with high horizontal wavenumbers. The D, CD, and other grids do not share this benefit from the nonhydrostatic effects because their errors are dominated by the averaging or computational modes at the SHRS. Nonhydrostatic effects do not produce similar benefits for the vertical discretization

because the benefits are mostly confined to the deep modes, which are already well resolved in both the nonhydrostatic and quasi-hydrostatic discrete systems.

## 8. Summary and conclusions

We have discussed the horizontal and vertical discretizations of the three-dimensional nonhydrostatic linearized anelastic equations on the A, B, C, CD, (DC), D, E, and Z horizontal grids and the L and CP vertical grids, with an emphasis on

middle-latitude inertia-gravity waves. The use of a three-dimensional nonhydrostatic system instead of the two-dimensional shallow-water system allows us to directly assess the performance of the discretization as a function of the vertical wavenumber, vertical resolution or vertical grid spacing.

The Z grid yields the most accurate inertia-gravity wave dispersion among the seven horizontal grids considered in this paper. It has no computational or dynamically inert modes. Although the frequency and group velocity of inertia-gravity

modes in the Z-grid solutions are lower than the true ones, the numerical frequency never goes below the inertial frequency and the group velocity never reverses sign.

The C grid produces mixed results for the inertia-gravity solutions. For cloud-resolving applications with small horizontal grid spacing represented by $d = 2$ km in our analyses, the accuracy of the physical modes is nearly identical to that of the physical modes with the Z-grid. Of course, there is a dynamically inert mode that decouples the divergence and vorticity for



the SRHS in the C-grid solution, but its impact may not be severe because it disperses like a pure gravity mode. However, since the three-dimensional enstrophy cascades to the SRHS in nonlinear cloud-resolving models, the impact of the dynamically inert mode can be severe, and a parameterized dissipation is needed to dissipate the enstrophy accumulated at the SRHS. For mesoscale applications, i.e., horizontal grid spacings in the range 10 km to 25 km, the inertia-gravity modes

of the C-grid solution behave like those of the Z-grid solutions if the vertical wavenumbers are equal to $n = 320$ or smaller for 10 km grid spacings, and $n = 156$ or smaller 25 km grid spacings. With a domain height of $z_T = 80$ km, wavenumbers $n = 320$ and $n = 156$ correspond to $\delta z = 250$ m and $\delta z = 512$ m, respectively. The inertia-gravity modes in the C-grid solutions with a typical climate model horizontal grid spacing of 100 km, are as accurate as the Z-grid solutions for vertical wave numbers $n = 39$ and below, which corresponds to $\delta z = 2051$ m and larger. For vertical wave numbers higher than

$n = 39$, the reversal of group velocity takes place for the modes over a rather wide range of horizontal scales. This might lead to noise or instability.

The inertia-gravity mode solutions with the D and CD grids are almost identical. On these grids, the divergence and the mass (buoyancy, pressure and vertical velocity) are placed at different grid points. The vorticity is placed at the same grid points with the mass. With this staggering, not only are the vorticity and divergence averaged to each other's grid points, but

also the divergence is averaged to mass (and vertical velocity) points and the pressure is averaged to divergence points. The result is large errors in the dispersion of the inertia-gravity modes for all vertical scales near the SRHS, for all horizontal grid spacings. At the SRHS, there are dynamically inert modes in all variables. The group velocity reverses and becomes large near the SRHS. In nonlinear models, the D and CD grids may need explicit diffusion to remove the noise in short horizontal scales that results from the computational mode and the reversal of the group velocity.

The dispersions of the inertia-gravity waves with the A, B and E grids are similar. All suffer from the existence of multiple (or non-unique) physical solutions that do not interact with each other (in linear models), as demonstrated by the numerical simulations. The existence of the multiple solutions with these grids also leads to the dynamically inert modes at the SRHS. To avoid the separation of two (or four) solutions with these grids, a horizontal mixing process is needed.

The conclusions of our normal-mode analyses of the nonhydrostatic anelastic inertia-gravity waves with the Z, C, D, CD,

A, E and B grids for the low horizontal wavenumbers is consistent with earlier the shallow-water analyses by various authors. The Z and C grids produce overall smaller errors with the nonhydrostatic anelastic system than with the quasi-hydrostatic system, particularly for the deep modes with high horizontal wavenumbers.

Part II (Konor and Randall, 2017b) discusses the dispersion of the middle-latitude Rossby waves all the horizontal grids.

*Competing interests.* The authors declare that they have no conflict of interest.

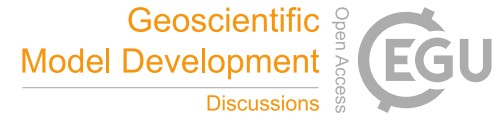

*Code and data availability.* Fortran and IDL codes that are used to compute and plot the frequencies for the CD grid are provided in http://doi.org/10.5281/zenodo.1117930. The codes that are used for the numerical inertia-gravity wave simulations with the CD grid are also provided at the same address.

*Acknowledgments.* We are grateful to Dr. Bill Skamarock for his comments and suggestions to improve the manuscript. This

research was supported by the NSF under AGS-1500187, the US Department of Energy Office of Science DE-SC07050 (SciDAC), DE-SC00016273 (ACME), and DE-SC00016305 (CMDV).

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



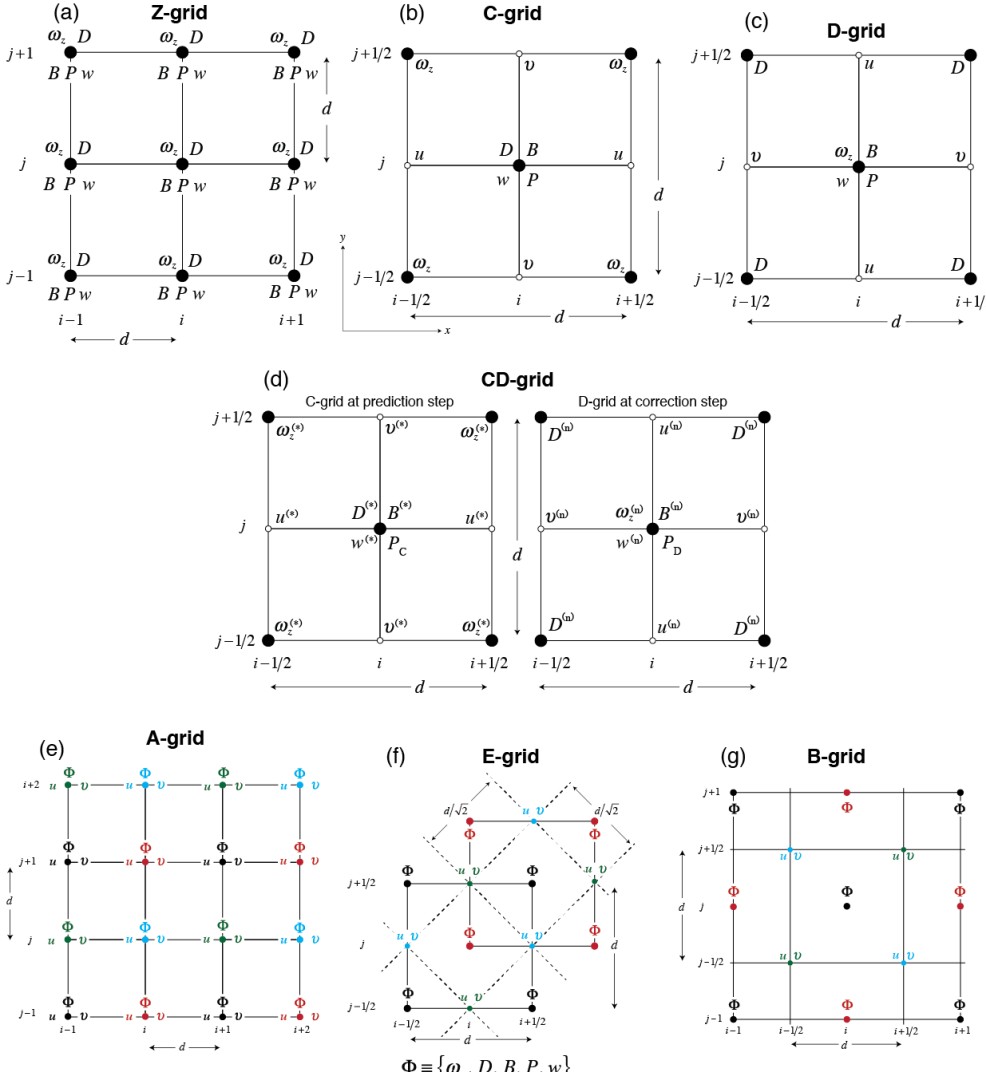

**Figure 1:** Portions of (a) the Z, (b) C, (c) D, (d) CD, (e) A, (f) E, and (g) B grids on a square grid. Colored grid points in A, E and B grids indicate the network of grid points, on which the decoupled solutions exist.





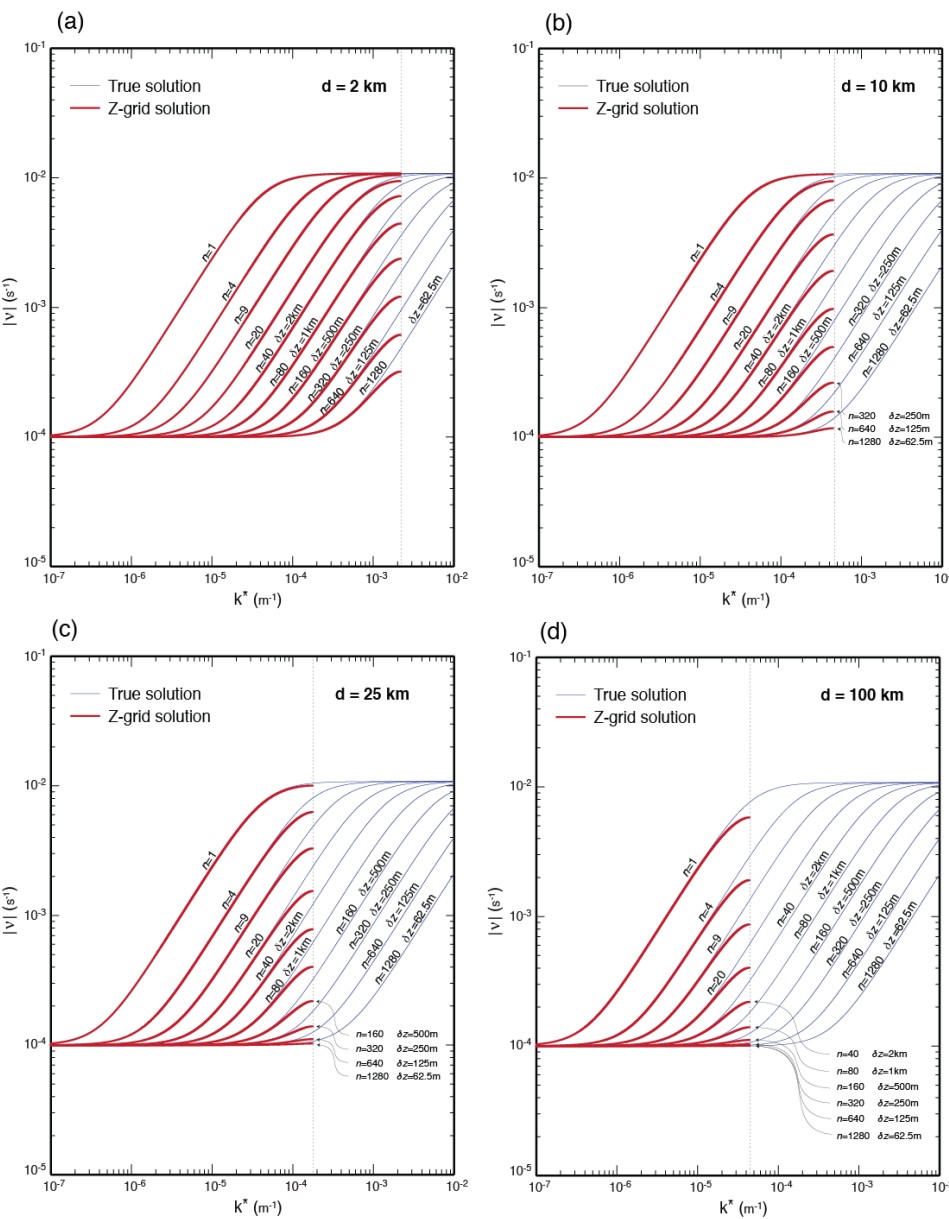

**Figure 2:** Plots of the absolute value of frequency obtained on the Z grid (red lines) for the grid spacings (a) 2 km, (b) 10 km, (c) 25 km, and (d) 100 km, and for the various vertical wave numbers. The blue thin lines are the corresponding true frequencies.



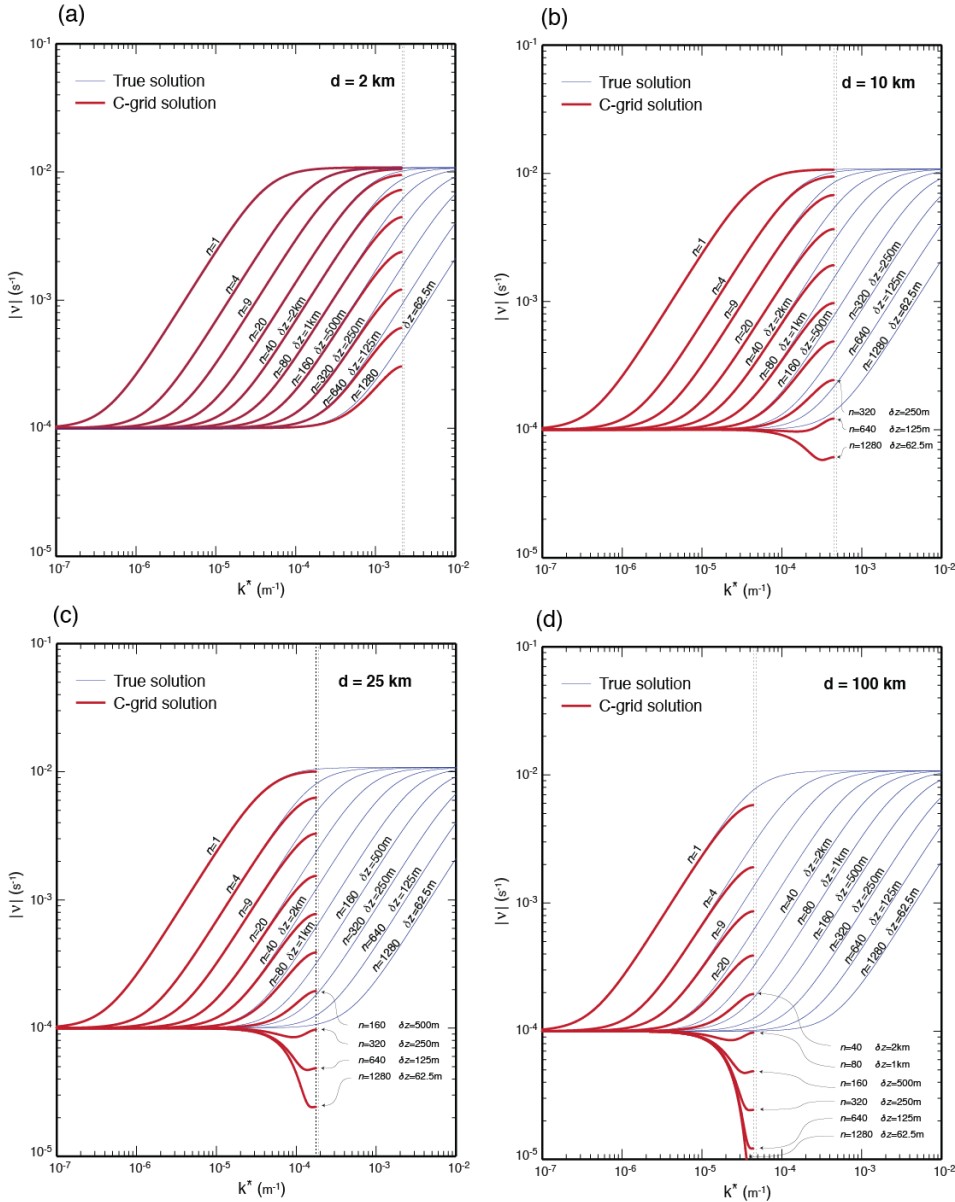

**Figure 3:** Same as Fig. 2, but for the C grid.



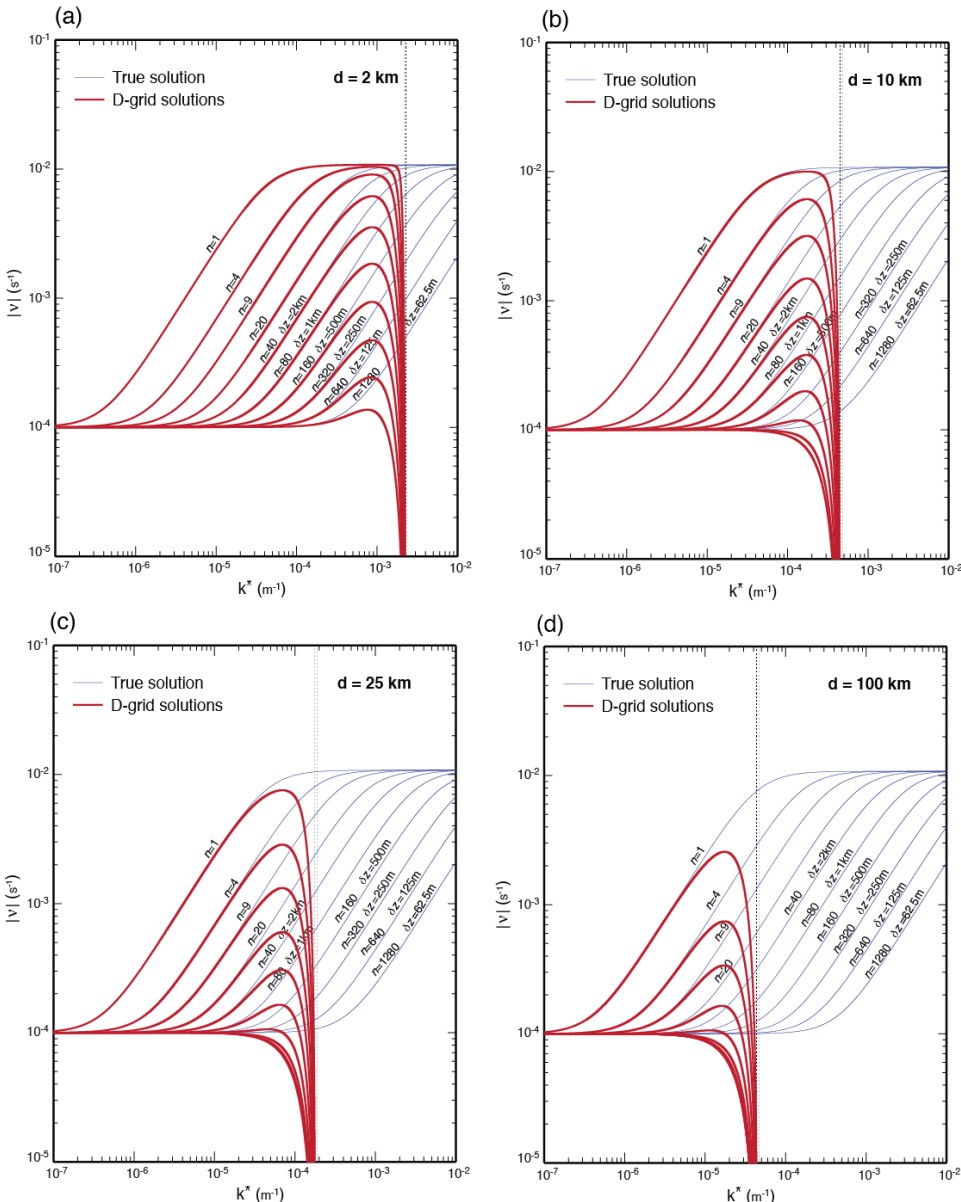

**Figure 4:** Same as Fig. 2, but for the D grid.





| Scheme I | Scheme II | Scheme III | Scheme IV | Scheme V |
|---|---|---|---|---|
| $\hat{P} \equiv \hat{P}_C = \hat{P}_D$ | $\hat{P}_C$ and $\hat{P}_D$ are treated separately | $\hat{P}_C$ and $\hat{P}_D$ are treated separately | $\hat{P}_C$ and $\hat{P}_D$ are treated separately | $\hat{P}_C$ and $\hat{P}_D$ are treated separately |
| | | $f\hat{\omega}_z^{(*)}$ is replaced by $\mu f\hat{\omega}_z^{(n+1)}$ in (39) | $\hat{B}^{(n)}$ is replaced by $\hat{B}^{(*)}$ in (34) | $f\hat{\omega}_z^{(*)}$ is replaced by $\mu f\hat{\omega}_z^{(n+1)}$ in (39) |
| | | $\hat{B}^{(*)}$ is replaced by $\hat{B}^{(n+1)}$ in (40) | $\hat{B}^{(*)}$ is replaced by $\hat{B}^{(n+1)}$ in (40) | $\hat{B}^{(*)}$ is replaced by $\hat{B}^{(n+1)}$ in (40) |
| | | | $\hat{w}^{(*)}$ is replaced by $\hat{w}^{(n+1)}$ in (41) | $f\hat{\omega}_z^{(n)}$ is replaced by $\mu f\hat{\omega}_z^{(*)}$ in (33) |

**Table 1:** A summary of the five schemes that we constructed using the CD-grid.





**Figure 5:** Same as Fig. 2, but for the CD grid using Scheme I.







**Figure 6:** Computational mode patterns for the A, B and E grids.



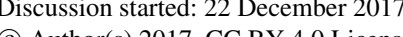

**Figure 7:** Same as Fig. 2, but for the A grid.



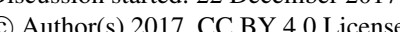


**Figure 8:** Same as Fig. 2, but for the E grid.





**Figure 9:** Same as Fig. 2, but for the B grid.




L-grid

CP-grid

———————— $w$ ————————     ———————— $w$   $B$ ———————— $k+3/2$

- - - - - $u$ $v$ $\omega_z$ $D$ $P$ $B$ - - - - - -     - - - - - - - $u$ $v$ $\omega_z$ $D$ $P$ - - - - - - - - $k+1$

———————— $w$ ————————     ———————— $w$   $B$ ———————— $k+1/2$

- - - - - $u$ $v$ $\omega_z$ $D$ $P$ $B$ - - - - - -     - - - - - - - $u$ $v$ $\omega_z$ $D$ $P$ - - - - - - - - $k$

———————— $w$ ————————     ———————— $w$   $B$ ———————— $k-1/2$

**Figure 10:** The L (left panel) and CP (right panel) grids and the distribution of variables.





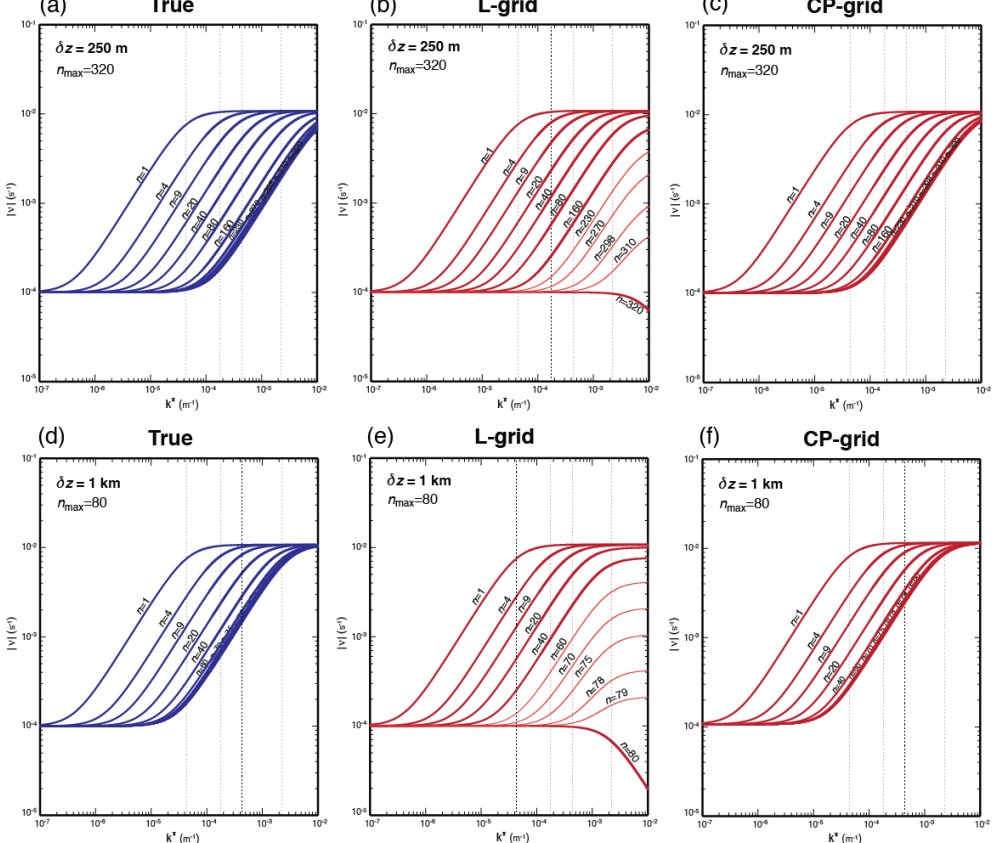

**Figure 11:** Plots of (a and d) true frequencies and discrete frequencies for (b and e) the L and (c and f) CP grids. The upper and lower panels show the plots for the maximum vertical integer wave numbers of $n_{max} = 320$ ( $\delta z = 250$ m ) and $n_{max} = 80$ ( $\delta z = 1$ km ), respectively.



| True | Z-grid |
|---|---|
| $$v^2 = \frac{N^2\left(k^2+\ell^2\right)+f^2\left(m^2+\frac{1}{4H^2}\right)}{\left(k^2+\ell^2\right)+\left(m^2+\frac{1}{4H^2}\right)}$$ | $$v^2 = \frac{N^2\left(\xi^2 k^2+\eta^2\ell^2\right)+f^2\left(m^2+\frac{1}{4H^2}\right)}{\left(\xi^2 k^2+\eta^2\ell^2\right)+\left(m^2+\frac{1}{4H^2}\right)} \quad 0\le[kd,\ \ell d]\le\pi$$ |
| $m \equiv \pi n/z_T \quad \text{for} \quad n=1,2,3,\cdots$ | $\xi \equiv \sin\left(\tfrac{1}{2}kd\right)/\left(\tfrac{1}{2}kd\right) \quad \text{and} \quad \eta \equiv \sin\left(\tfrac{1}{2}\ell d\right)/\left(\tfrac{1}{2}\ell d\right)$ |

| C-grid | D-grid |
|---|---|
| $$v^2 = \frac{N^2\left(\xi^2 k^2+\eta^2\ell^2\right)+\mu^2 f^2\left(m^2+\frac{1}{4H^2}\right)}{\left(\xi^2 k^2+\eta^2\ell^2\right)+\left(m^2+\frac{1}{4H^2}\right)}$$ | $$v^2 = \frac{\mu^2 N^2\left(\xi^2 k^2+\eta^2\ell^2\right)+\mu^2 f^2\left(m^2+\frac{1}{4H^2}\right)}{\mu^2\left(\xi^2 k^2+\eta^2\ell^2\right)+\left(m^2+\frac{1}{4H^2}\right)}$$ |
| $\mu \equiv \cos\left(\tfrac{1}{2}kd\right)\cos\left(\tfrac{1}{2}\ell d\right) \quad 0\le[kd,\ \ell d]\le\pi$ | $0\le[kd,\ \ell d]\le\pi$ |

| CD-grid (Scheme I) |
|---|
| $$e^{2v_r\tau}\left(\mu^2 L^2+\sigma_m^2\right)\cos\left(2v_r\tau\right)-2e^{v_r\tau}\left(\mu^2\sigma_N L^2+\sigma_f\sigma_m^2\right)\cos\left(v_r\tau\right) \qquad e^{v_r\tau}=\frac{2\left(\mu^2\sigma_N L^2+\sigma_f\sigma_m^2\right)}{\left(\mu^2 L^2+\sigma_m^2\right)}\frac{\sin\left(v_r\tau\right)}{\sin\left(2v_r\tau\right)}$$ |
| $$+\left(\sigma_N^2+\tau^2 N^2\right)\mu^2 L^2+\left(\sigma_f^2+\tau^2\mu^2 f^2\right)\sigma_m^2=0$$ |
| $\sigma_N \equiv 1-\tfrac{1}{2}\tau^2 N^2 \qquad \sigma_f \equiv 1-\tfrac{1}{2}\tau^2 f^2 \qquad \sigma_m^2 \equiv m^2+1/\left(4H^2\right) \qquad 0\le[kd,\ \ell d]\le\pi$ |

| A grid | E grid |
|---|---|
| $$v^2 = \frac{N^2\left(\tilde\xi^2 k^2+\tilde\eta^2\ell^2\right)+f^2\left(m^2+\frac{1}{4H^2}\right)}{\left(\tilde\xi^2 k^2+\tilde\eta^2\ell^2\right)+\left(m^2+\frac{1}{4H^2}\right)} \quad 0\le[kd,\ \ell d]\le\pi$$ | $$v^2 = \frac{N^2\left(\xi^2 k^2+\eta^2\ell^2\right)+f^2\left(m^2+\frac{1}{4H^2}\right)}{\left(\xi^2 k^2+\eta^2\ell^2\right)+\left(m^2+\frac{1}{4H^2}\right)}$$ |
| $\tilde\xi \equiv \sin\left(kd\right)/\left(kd\right) \quad \text{and} \quad \tilde\eta \equiv \sin\left(\ell d\right)/\left(\ell d\right)$ | $0\le[kd,\ \ell d]\le 2\pi$ |

| B grid |
|---|
| $$v^2 = \frac{N^2\left(\xi^2 k^2+\eta^2\ell^2-\frac{1}{2}d^2\xi^2 k^2\eta^2\ell^2\right)+f^2\left(m^2+\frac{1}{4H^2}\right)}{\left(\xi^2 k^2+\eta^2\ell^2-\frac{1}{2}d^2\xi^2 k^2\eta^2\ell^2\right)+\left(m^2+\frac{1}{4H^2}\right)} \quad 0\le[kd,\ \ell d]\le\pi$$ |

| L grid | CP grid |
|---|---|
| $$v^2 = \frac{\mu_z^2 N^2\left(k^2+\ell^2\right)+f^2\left(\zeta^2 m^2+\mu_z^2\frac{1}{4H^2}\right)}{\left(k^2+\ell^2\right)+\left(\zeta^2 m^2+\mu_z^2\frac{1}{4H^2}\right)}$$ | $$v^2 = \frac{N^2\left(k^2+\ell^2\right)+f^2\left(\zeta^2 m^2+\mu_z^2\frac{1}{4H^2}\right)}{\left(k^2+\ell^2\right)+\left(\zeta^2 m^2+\mu_z^2\frac{1}{4H^2}\right)}$$ |
| $\zeta \equiv \sin\left(\tfrac{1}{2}m\delta z\right)/\left(\tfrac{1}{2}m\delta z\right) \qquad \mu_z \equiv \cos\left(\tfrac{1}{2}m\delta z\right) \qquad 0\le m\delta z=\pi n\delta z/z_T\le\pi \quad \text{for} \quad n=1,2,3,\cdots$ | |

**Table 2:** A summary of the true and discrete dispersion relations for the horizontal and vertical grids discussed in this paper.



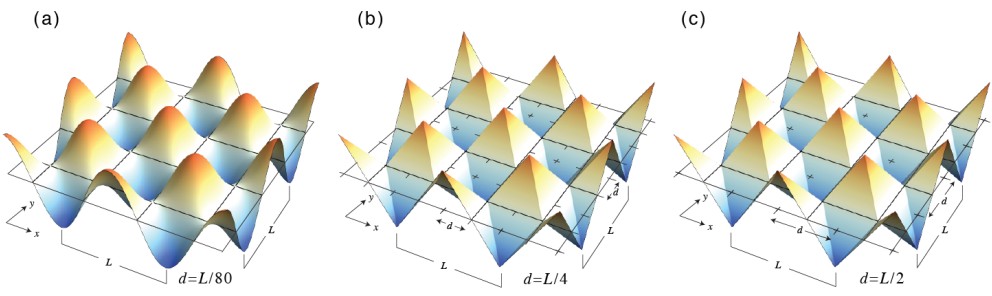

**Figure 12:** Initial perturbations used in the simulations with the Z, C, CD and D grid models for the grid spacings of (a) $d = L/80$, (b) $d = L/4$, and (c) $d = L/2$.





| Horizontal Scale of Perturbation | Vertical wavenumber | True frequency (analytic) $\tilde{B} \quad D \quad \omega_z$ | Horizontal grid distance (d) | Numerical frequency | | | |
|---|---|---|---|---|---|---|---|
| | | | | Z grid $\tilde{B} \quad D \quad \omega_z$ | C-grid $\tilde{B} \quad D \quad \omega_z$ | CD-grid $\tilde{B} \quad D \quad \omega_z$ | D-grid $\tilde{B} \quad D \quad \omega_z$ |
| | $n$=320 | 18.84724224 | | 18.84578676 | 18.84578676 | 18.81193205 | 18.81193205 |
| $L$ = 4 km | $n$=640 | 9.57153193 | 50 m | 9.56927399 | 9.56927399 | 9.55472218 | 9.55472218 |
| | $n$=1280 | 4.87709384 | | 4.87597804 | 4.87522137 | 4.86842190 | 4.86842190 |
| | $n$=80 | 1.82682191 | | 1.82650735 | 1.82565821 | 1.82364465 | 1.82364465 |
| $L$ = 200 km | $n$=160 | 1.25874004 | 2.5 km | 1.25860047 | 1.25739149 | 1.25668732 | 1.25668732 |
| | $n$=320 | 1.07056681 | | 1.07053521 | 1.06907801 | 1.06885977 | 1.06885977 |

**Table 3:** True and numerically obtained frequencies from the simulations for the Z, C, CD and D grids using a horizontal grid spacing of $d = L/80$. The unit for the frequency is $10^{-4}\,\text{s}^{-1}$. The horizontal wavenumber for these cases can be obtained from $2\pi/L$.



| Horizontal Scale of Perturbation | Vertical wavenumber | True frequency (analytic) $\tilde{B}\ D\ \omega_z$ | Horizontal grid distance (d) | Numerical frequency | | | |
|---|---|---|---|---|---|---|---|
| | | | | Z grid $\tilde{B}\ D\ \omega_z$ | C-grid $\tilde{B}\ D\ \omega_z$ | CD-grid $\tilde{B}\ D\ \omega_z$ | D-grid $\tilde{B}\ D\ \omega_z$ |
| | $n$=320 | 18.84724224 | | 17.01837840 | 16.99996024 | 8.51149459 | 8.51149459 |
| $L$ = 4 km | $n$=640 | 9.57153193 | 1 km | 8.63549382 | 8.59062798 | 4.31715357 | 4.31715357 |
| | $n$=1280 | 4.87709384 | | 4.41296903 | 4.32726260 | 2.20663949 | 2.20663949 |
| | $n$=80 | 1.82682191 | | 1.70137701 | 1.46447541 | 0.85068850 | 0.85066547 |
| $L$ = 200 km | $n$=160 | 1.25874004 | 50 km | 1.21395442 | 0.85073457 | 0.60700066 | 0.60698893 |
| | $n$=320 | 1.07056681 | | 1.05756165 | 0.60700066 | 0.52879862 | 0.52878082 |

**Table 4:** Same as Table 2, but using $d = L/4$. Underlined numbers indicate frequencies less than the minimum frequency of inertial oscillation ($10^{-4}\,\mathrm{s}^{-1}$).

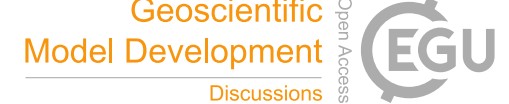

| Horizontal Scale of Perturbation | Vertical wavenumber | True frequency (analytic) $\tilde{B}\ D\ \omega_z$ | Horizontal grid distance ($d$) | Numerical frequency | | | | |
|---|---|---|---|---|---|---|---|---|
| | | | | Z grid $\tilde{B}\ D\ \omega_z$ | C-grid $\tilde{B}\ \ D$ | $\omega_z$ | CD-grid $\tilde{B}\ D\ \omega_z$ | D-grid $\tilde{B}\ D\ \omega_z$ |
| | $n$=320 | 18.84724224 | | 12.13438645 | 12.09235047 (0.0) | 0.0 | 0.0 (0.0) | 0.0 (0.0) |
| $L$ = 4 km | $n$=640 | 9.57153193 | 2 km | 6.15636420 | 6.07423173 (0.0) | 0.0 | 0.0 (0.0) | 0.0 (0.0) |
| | $n$=1280 | 4.87709384 | | 3.20070859 | 3.04064329 (0.0) | 0.0 | 0.0 (0.0) | 0.0 (0.0) |
| | $n$=80 | 1.82682191 | | 1.39545713 | 0.97338269 (0.0) | 0.0 | 0.0 (0.0) | 0.0 (0.0) |
| $L$ = 200 km | $n$=160 | 1.25874004 | 100 km | 1.21395442 | 0.48670642 (0.0) | 0.0 | 0.0 (0.0) | 0.0 (0.0) |
| | $n$=320 | 1.07056681 | | 1.05756165 | 0.24335698 (0.0) | 0.0 | 0.0 (0.0) | 0.0 (0.0) |

**Table 5:** Same as Table 4, but using $d = L/2$. Zero frequencies indicate dynamically inert computational modes. Zeros in the parenthesis indicate the frequency of the buoyancy and divergence if the initial perturbation is only given to the vorticity.



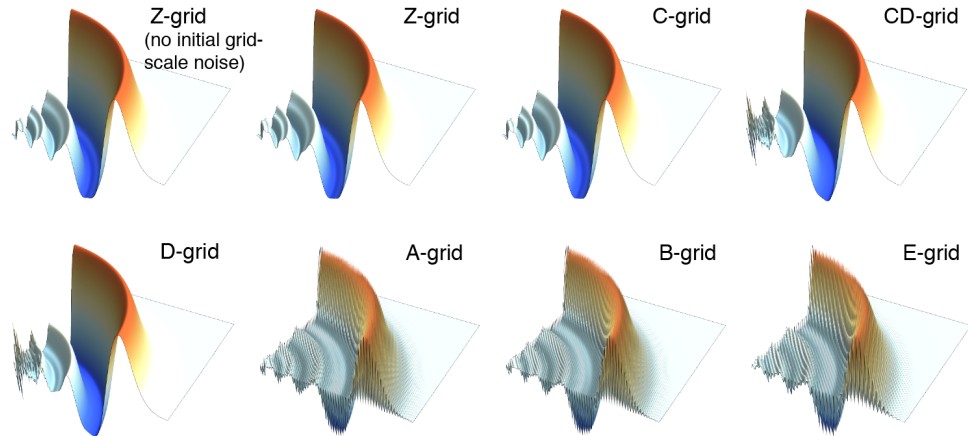

**Figure 13:** Horizontal structure of the inertia-gravity wave simulated by the various horizontal grids after 100 mins of integration.





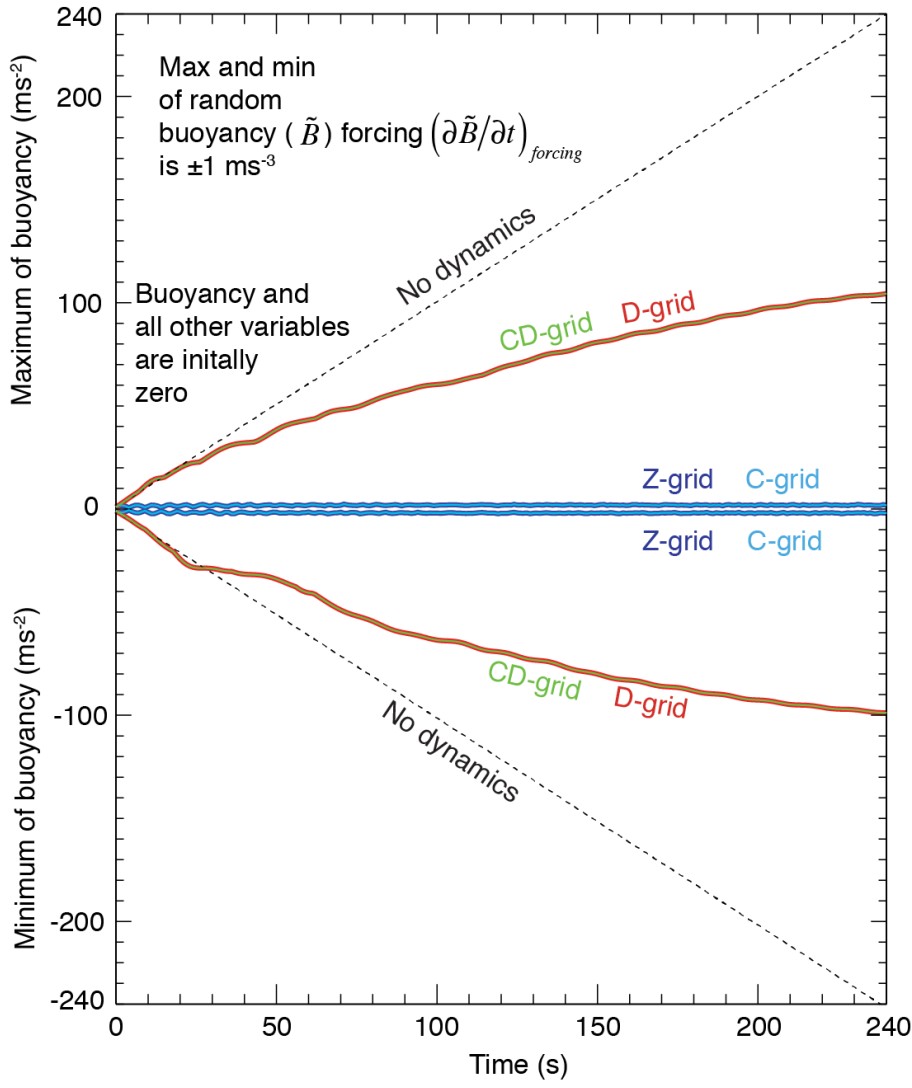

**Figure 14:** Time change of the maxima and minima of buoyancy ($\hat{B}$) for the Z-, C-, D-, and DC-grid simulations in the presence of noisy heating patterns. The thin dashed line indicates the time change when the response of vorticity and divergence is turned off.





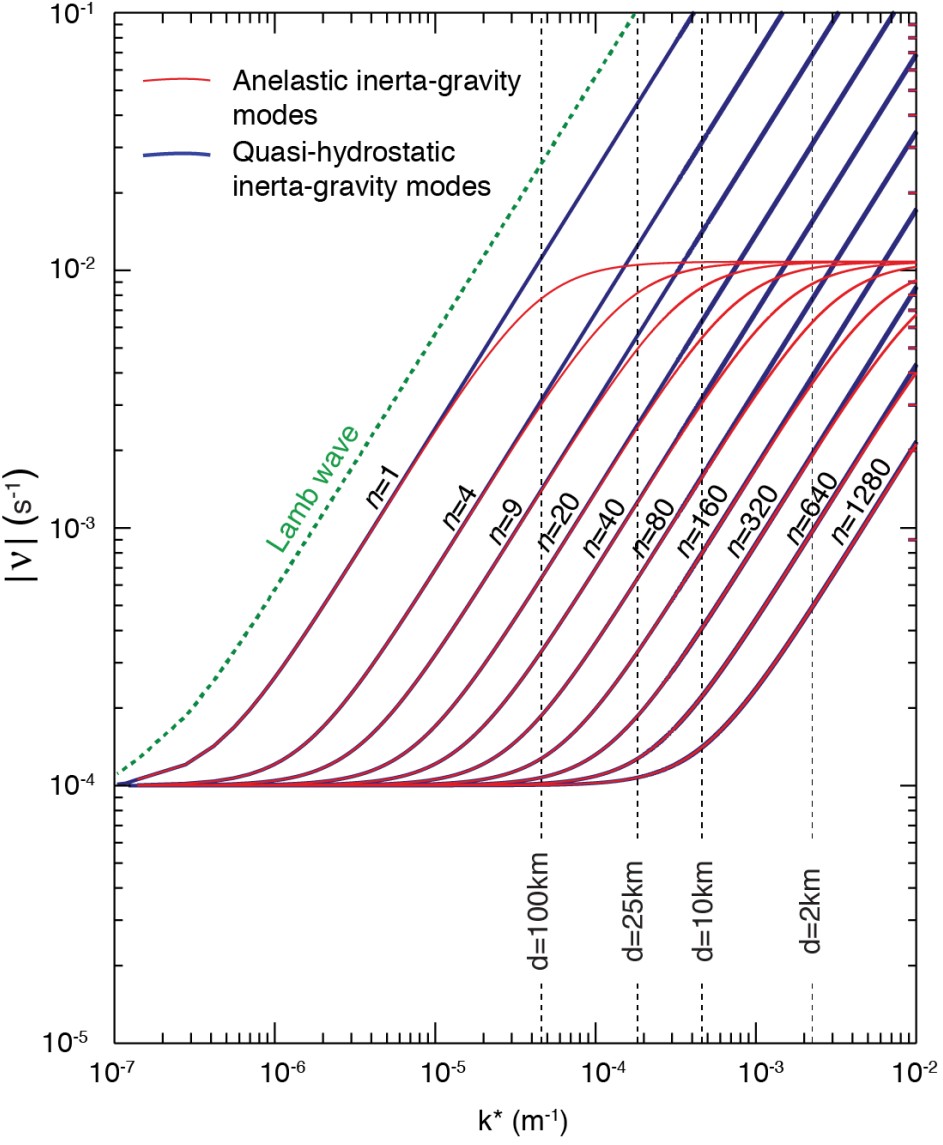

**Figure 15:** Plots of the absolute value of frequency for inertia-gravity modes obtained with the anelastic (red curves) and the quasi-hydrostatic (black curves) systems. The frequency of Lamb wave with the quasi-hydrostatic system is indicated with the dashed green curves.