# Peer review of "Impacts of the Horizontal and Vertical Grids on the Numerical Solutions of the Dynamical Equations. Part I: Nonhydrostatic Inertia-Gravity Modes"

_Geoscientific Model Development, 2017_

## Short Comment (SC1) · 9 Jan 2018

Comments on Konor and Randall

by Lucas Harris and Xi Chen

This manuscript makes broad conclusions regarding grid staggering (which is merely one characteristic out of the many of a modern dynamical core) based on the analysis of a simplified, linear, centered-difference, second-order discretization. We feel that the analysis presented in the manuscript is unrepresentative of the discretizations used in

modern dynamical cores, and that the conclusions may be misleading as to the actual behavior of a discretization in a comprehensive numerical model.

The authors did a remarkable job by acquiring analytical solutions or workably simple numerical solutions to the two-dimensional linearized analysis, but the mathematics is made tractable only by analyzing an oversimplified second-order centered-difference system that greatly exaggerates the difference between the staggerings. Few modern dynamical cores use such a primitive method; most use at least third or fourth-order, and many finite-volume schemes use a physically-based upwinding method. The analysis method also necessarily neglects many important fluid phenomena, especially nonlinear vorticity advection, crucial to many of the uses of numerical models.

Further, the analysis is entirely inviscid, while all numerical models require either implicit or explicit diffusion to remove grid-scale noise, created by improper handling of sharp gradients or errors in the parameterizations, boundaries, and data. Indeed, at $2\Delta x$ it is nearly impossible to distinguish computational noise from physical signals, so these modes should be filtered out (Skamarock et al. 2014). To this end, the behavior of the staggerings at $2\Delta x$ and $3\Delta x$—which the authors' conclusions lie heavily upon— is of little consequence. We can also conclude that the numerical diffusion, whether implicit or explicit, will always be an intrinsic part of any dynamical core, and a thorough analysis of any discretization must also consider the effects of dissipation.

Finally, the C-D grid analyzed in the manuscript has little resemblance to that in the FV3 dynamical core. The discretization is described in detail in Lin and Rood (1997), Lin (2004), and Harris and Lin (2013), and which can also be seen in the FV and FV3 source code that has been openly available for many years in GEOS, CESM, and the GFDL modeling suite. In FV3 none of the time-advanced C-grid quantities, other than the winds, are used for the full forward timestep. The C-grid vorticity, temperature, and mass are discarded, in contradiction to eqns. (39) through (42); this incorrect procedure, which then necessitates additional averaging of the vorticity to cell-centers, introduces significant error. FV3 also implements a fourth-order accurate transport

scheme and uses upwinding based on the Lin and Rood (1996) advection scheme, and so would have behavior greatly different than that of the second-order method used in the manuscript, even if the analysis was done consistently with the FV3 algorithm.

References

Harris, L.M. and S. Lin, 2013: A Two-Way Nested Global-Regional Dynamical Core on the Cubed-Sphere Grid. Mon. Wea. Rev., 141, 283–306

Lin, S., 2004: A "Vertically Lagrangian" Finite-Volume Dynamical Core for Global Models. Mon. Wea. Rev., 132, 2293–2307

Lin, S. and R.B. Rood, 1996: Multidimensional Flux-Form Semi-Lagrangian Transport Schemes. Mon. Wea. Rev., 124, 2046–2070

Lin, S.-J. and R.B. Rood, 1997: An explicit flux-form semi-lagrangian shallow-water model on the sphere. Q.J.R. Meteorol. Soc., 123, 2477–2498.

Skamarock, W.C., S. Park, J.B. Klemp, and C. Snyder, 2014: Atmospheric Kinetic Energy Spectra from Global High-Resolution Nonhydrostatic Simulations. J. Atmos. Sci., 71, 4369–4381

---

## Referee Comment (RC1) · A. Gaßmann (Referee) · 11 Jan 2018

I really appreciated the profound analyses of the very comprehensive paper. At least to my knowledge, nobody before had made such a synoptic analysis of the diverse grid staggerings for the nonhydrostatic equations. In this respect I learned how the nonhydrostatic wave dispersion properties differ from the gravity-wave dispersion properties with respect to the chosen grid staggering, an issue which I never thought about before.

When developing dynamical cores, the choice of grid staggering or – in newer cores – the choice of base functions (e.g. which order of spectral elements) stands at the begin of the work. The sequence of such choices goes – like in many philosophical questions – from simple to complicated. For the dynamical core development the question of the linear regime must be answered first, before posing questions about nonlinearity and energy cascades in a second step. In that regard I feel that the other online comment of Harris and Chen is inadequate and only focused on the direct defense of their work. As I read the manuscript, I can see that the authors aim very much on scientific neutrality – as is a must for a honest author. They do not conceal that a Z-grid needs 10-20\% overhead due to the required inversions and they try several different versions of the CD-grid with the goal finding the most appropriate. If I was the other reviewer and I would see that the authors have not programmed what I know, I would simply write down the scheme and require the analysis in a revision.

---

## Referee Comment (RC2) · A. Gaßmann (Referee) · 11 Jan 2018

Sorry, I pushed the submit button before I was finishing.

I only have a few comments on the manuscript, which are not difficult to discuss.

- What would change if the stratification is not isothermal? The isothermal state is not representative to the real atmosphere.

- I wonder how the well known fact that vertical and horizontal grid spacing must

be in a way consistent for an atmospheric model as found by Lindzen and Fox-Rabinovitz (1989) might be seen by such a 3-dimensional analysis.

- The mentioning of the hexagonal C-grid in lines 17 at page 10 is perhaps not adequate at this place because you discuss the quadrilateral C-grid. The hexagonal C-grid problems are due to the overspecification of the horizontal wind components. This issue occurs additionally to the inspection of usual wave dispersion relations, and has no direct relation to the C-grid staggering philosophy. When guaranteeing the linear dependence of all forcing terms in the momentum equations, the dispersion relation on the hexagonal C-grid is very similar to that on the quadrilateral grid. The problem with the triangular C-grid is that the linear dependence can never be achieved, whereas on the hexagonal C-grid, the linear dependence can be achieved. Therefore the triangular C-grid needs diffusion or divergence averaging for numerical reasons, whereas the the diffusion on the hexagonal C-grid might be more interpreted in a context of the description of the physically consistent energy cascade.

I like the paper very much, I think it is suitable for educational purposes.

---

## Referee Comment (RC3) · Anonymous Referee #2 · 16 Jan 2018

Referee comment on "Impacts of the Horizontal and Vertical Grids on the Numerical Solutions of the Dynamical Equations. Part I: Nonhydrostatic Inertia-Gravity Modes" by C.S. Konor and D.A. Randall

The manuscript derives and discusses numerical dispersion relationships for finite difference approximations of the Lipps & Hemler anelastic equations, for various staggered arrangements of the variables. In part I, the analysis is done on a mid-latitude $f-$plane and focuses on inertia-gravity modes ; in part II the analysis is done on a mid-latitude $\beta-$plane and focuses on Rossby mode.

Such numerical dispersion analysis are useful to understand the behavior of numerical schemes near the grid scale. Although one can not expect any numerical scheme to be accurate at such scales, one hopes to avoid pathological behavior such as physically propagating modes being numerically stationary or vice-versa. The alternative is to apply a sufficiently large diffusion that damps the small-scale degrees of freedom. This waste of degrees of freedom can be deemed acceptable in exchange of a gain in simplicity (e.g. collocated A-grid) in certain cases, e.g. spectral methods with scale-selective high-order hyperviscosity, but would be much more disputable for staggered lowish-order finite-difference / finite-volume methods. That such linear analyses are useful is demonstrated by the long history of similar work, and the operational popularity of staggered-mesh methods or, more recently, compatible finite element methods.

Compared to previous work, the work presented here is novel in 3 ways : (a) it deals with three-dimensional, non-hydrostatic equations while previous work two-dimensional shallow-water equations or hydrostatic equations that are equivalent to the latter after separation of variables (e.g. Bell, Peixoto, Thuburn QJRMS 2017) (b) it adresses the C-D staggering, in addition to better understood staggerings (A-E, Z) (c) the analysis of the C-D staggering is done with discrete time, while the only existing analysis (Skamarock, MWR 2008) is for continuous time.

Anelastic equations rather than the fully compressible equations are analyzed. As discussed by the authors, it is plausible that despite some inaccuracies of anelastic equations this is not a serious limitation. Similarly, lowest-order centered finite-difference schemes are considered rather than higher-order, upwind biased schemes. This is not a serious limitation either, since (a) when linearizing about a state of rest, upwinding becomes irrelevant (b) increasing the order of a scheme with problematic dispersion properties only makes the problem worse as far as I am aware.

I was particularly interested by the analysis of the C-D grid. So far I have checked the math mostly of that part. As far as I understand Harris & Lin, 2013, the linearization (32-42) of the predictor-corrector time scheme is correct (notice that the horizontal curl

and div have been applied to the linearized momentum equations).

The final results refine as a function of horizontal scale previous wisdom accumulated on the pros and cons of the various staggerings. Especially it confirms that, with respect to numerical dispersion, the C-D grid is extremely similar to the D-grid, whose shortcomings with respect to the propagation of gravity waves are well-known. No breakthrough here, but a valuable addition of a missing piece to the puzzle.

Overall, this is a clearly a good paper, with a sound methodology and useful purposes, that deserves publication after possibly correcting minor issues (see separate comment).

---

## Editor Comment (EC1) · P. A. Ullrich (Editor) · 18 Jan 2018

Dear Drs. Harris + Chen,

Thank you for your short comment on this manuscript. I am very interested in resolving this apparent contradiction with regards to the analysis of the C-D grid. It has now been claimed by Anonymous Review #2 that the analysis is consistent with the formulation described in Harris and Lin (2013). In the interests of ensuring the correctness and consistency of the manuscript, it would be immensely helpful if you could point out the

revisions needed to equations (32)-(42) to render them consistent with the formulation in the FV3 dynamical core. This would be further helpful in ensuring no apparent misinformation about the C-D grid was propagated as a result of this analysis.

Your assistance on this matter would be greatly appreciated.

$\sim$ Paul Ullrich

---

## Author Comment (AC1) · 15 Mar 2018

We thank the referee for very instructive comments and going through the lengthy derivatives. We clarified in the text that Skamarock (2008) presents an analysis of the time continuous system.

---

## Author Comment (AC2) · 15 Mar 2018

We thank Dr. Gassmann for very useful comments.

Answer to comment (1):

We can think of two major effects of (vertically) varying basic-state stability expressed by the Brunt-Vaisala frequency.

1- Abrupt change of stability

This is the case with the tropopause. The abrupt change of stability at the tropopause can act like a lid for the troposphere. As a result, the vertical extent of the waves is limited. From this point of view, we can estimate that the longest realizable vertical wavelength is limited to, say, n=9 for the domain used in our analyses. Recall that n=9 corresponds to roughly a 9-km half-wavelength that fits in to the troposphere. The discussion in section 7 of our manuscript indicates that nonhydrostatic effects are more important for the deep modes (see Fig. 15). The absence of very deep waves in realistic situations may be the reason that quasi-hydrostatic models, in practice, can give good solutions with horizontal grid spacing as small as 10 km

2-Gradual change of stability

Gradual changing static stability can cause internal refractions and reflections of waves, which also may limit their vertical extent.

Answer to comment (2):

Lindzen&Fox-Robinovitz (1989) suggest that the maximum vertical grid spacing used to resolve the quasi-geostrophic (and quasi-static) modes in the presence of critical layers should satisfy [see Fig.1 for this equation]. In practice, this means that the vertical grid spacing should not exceed about 1% of the horizontal grid spacing. For nonhydrostatic models with small horizontal grid spacings, using such a small vertical grid spacing can immensely increase the computational cost. However, the nonhydrostatic models may not have to satisfy Lindzen&Fox-Robinovitz rule, because the resolutions that are needed to simulate nonhydrostatic motions may automatically resolve the QG modes.

We prepared the following table [see Fig.2 for this table], showing the horizontal and vertical propagation speeds of the waves found in our analysis. The table can used in the selection of the horizontal and vertical grid spacings. In making the table, we assumed that the deepest and fastest wave has n=8.
Answer to comment (3):

At the end of the C-grid subsection, we added a short paragraph discussing the C-grid discretization on the hexagonal and triangular grids (page 12, line 17, of revised Part I).

[Figure]

$$(\delta z)_{max} = (f/N)\delta x$$

**Fig. 1.** Equation

[Figure]

$m = \pi n / z_T$    $z_T = 80$ km

**Inertia-gravity waves**

| | $n$ | $k^* = \sqrt{2}\,\pi/d$ | $c_H \equiv v/k^*$ | $c_z \equiv v/m$ | $c_z/c_H$ |
|---|---|---|---|---|---|
| $d = 2$ km | 8 | 0.222144146908E-02 | 0.481901844357E+01 | 0.340756062011E+02 | 0.707106781187E+01 |
| | 9 | 0.222144146908E-02 | 0.480651989192E+01 | 0.302108694176E+02 | 0.628539361055E+01 |
| | 10 | 0.222144146908E-02 | 0.479266536687E+01 | 0.271114094470E+02 | 0.565685424949E+01 |
| $d = 10$ km | 8 | 0.444288293816E-03 | 0.198560405096E+02 | 0.280806817837E+02 | 0.141421356237E+01 |
| | 9 | 0.444288293816E-03 | 0.190326141538E+02 | 0.239254942789E+02 | 0.125707872211E+01 |
| | 10 | 0.444288293816E-03 | 0.182234775779E+02 | 0.206175113154E+02 | 0.113137084990E+01 |
| $d = 25$ km | 8 | 0.177715317526E-03 | 0.299096358343E+02 | 0.169194450570E+02 | 0.565685424949E+00 |
| | 9 | 0.177715317526E-03 | 0.272982045604E+02 | 0.137263968419E+02 | 0.502831488844E+00 |
| | 10 | 0.177715317526E-03 | 0.250598688090E+02 | 0.113408020291E+02 | 0.452548339959E+00 |
| $d = 100$ km | 8 | 0.444288293816E-04 | 0.340767827966E+02 | 0.481918483930E+01 | 0.141421356237E+00 |
| | 9 | 0.444288293816E-04 | 0.303836335749E+02 | 0.381946192673E+01 | 0.125707872211E+00 |
| | 10 | 0.444288293816E-04 | 0.274118268152E+02 | 0.310129418012E+01 | 0.113137084990E+00 |

**Rossby waves**

| | $n$ | $k^* = \sqrt{2}\,\pi/d$ | $c_H \equiv v/k^*$ | $c_z \equiv v/m$ | $c_z/c_H$ |
|---|---|---|---|---|---|
| $d = 2$ km | 8 | 0.222144146908E-02 | -0.232129064262E-05 | -0.164140035450E-04 | 0.707106781187E+01 |
| | 9 | 0.222144146908E-02 | -0.232128958774E-05 | -0.145902187430E-04 | 0.628539361055E+01 |
| | 10 | 0.222144146908E-02 | -0.232128840876E-05 | -0.131311901994E-04 | 0.565685424949E+01 |
| $d = 10$ km | 8 | 0.444288293816E-03 | -0.580298728932E-04 | -0.820666332683E-04 | 0.141421356237E+01 |
| | 9 | 0.444288293816E-03 | -0.580292136537E-04 | -0.729472897449E-04 | 0.125707872211E+01 |
| | 10 | 0.444288293816E-03 | -0.580284768744E-04 | -0.656517271998E-04 | 0.113137084990E+01 |
| $d = 25$ km | 8 | 0.177715317526E-03 | -0.362604929513E-03 | -0.205120323640E-03 | 0.565685424949E+00 |
| | 9 | 0.177715317526E-03 | -0.362579191118E-03 | -0.182316234494E-03 | 0.502831488844E+00 |
| | 10 | 0.177715317526E-03 | -0.362550429001E-03 | -0.164071594796E-03 | 0.452548339959E+00 |
| $d = 100$ km | 8 | 0.444288293816E-04 | -0.577841322363E-02 | -0.817191034986E-03 | 0.141421356237E+00 |
| | 9 | 0.444288293816E-04 | -0.577188385698E-02 | -0.725571238310E-03 | 0.125707872211E+00 |
| | 10 | 0.444288293816E-04 | -0.576460376944E-02 | -0.652190466596E-03 | 0.113137084990E+00 |

**Fig. 2.** Table

---

## Author Comment (AC3) · 15 Mar 2018

1-..."Unrepresentative of the discretizations used in modern dynamical cores"...:

We do not agree that the linear analyses presented in this paper are irrelevant for today's dynamical cores. The methods that grid-point models use to simulate wave propagation have not changed over the years.

2-..."Analyzing an oversimplifies second-order centered-difference system"...:

Gravity waves propagate horizontally in all directions with the same speed, so the horizontal discretization of terms responsible for gravity wave propagation should be based on a centered treatment. Since our main concern is the behavior of the solutions near the smallest resolved horizontal scale (SRHS), we used a second-order scheme. Higher-order schemes produce more accurate solutions of the well-resolved features, but low-order schemes can actually be more accurate near the SRHS. We comment on the potential impact of higher-order schemes in section 7c below.

3-..."Analysis method neglects nonlinear vorticity advection"...:

Nonlinear processes are important to control the spurious cascades of (potential) enstrophy and kinetic energy to small scales, and they limit the accumulation of noise near the SRHS. They act slowly, however, and should not be expected to "cure" the rapid adverse-effects of poorly simulated linear wave dispersion.

When parameterized physical processes and topography are included, noise can be generated even without a spurious cascade. In such a case, wave propagation on the smallest resolved scales can disperse the energy and thereby reduce the noise. Of course, diffusion can also help to dissipate the noise, but a poor scheme may require excessive diffusion that also damps some of the better-resolved scales, thus effectively reducing the resolution of the model.

4-..."The analysis is entirely inviscid"...:

Yes, our analysis is entirely inviscid, and it should be. Viscous effects do not affect the dispersion of waves, but simply reduce their amplitudes. Avoiding the need for excessive diffusion is an important example of "good practices in dynamical core development."

5-..." The CD grid analyzed has little resemblance to that used in the FV3"...:

We gave the reviewers every chance to tell us what their scheme is. We personally asked them twice. In response, they gave some references, which describe the

scheme in vague terms, without equations. The Editor asked the reviewers to specify the changes that would be needed to make equations (32)-(42) consistent with the formulation of FV3. The reviewers have not responded.

We have analyzed more than a dozen possible CD schemes, some of which have not produced closed systems of equations. (We will explain what we mean by closed systems in item 7a below.) All of the schemes that produced closed systems have been analyzed and presented in our manuscript and the supplementary material.

6-..." The C grid vorticity temperature and mass discarded"...:

Schemes III and V (pages 57 and 70 of the supplementary material, respectively) discard the vorticity obtained on the C grid, and pass the divergence to the D grid. At least in the shallow water sense, these schemes mimic what the reviewers describe as their scheme. The frequencies for scheme III and V are shown in figures on page 62 and page 75 of the supplementary material, respectively. The frequencies shown in these figures are virtually identical to the frequencies for scheme I that is discussed in the manuscript and all other schemes discretized on the CD grid.

All of the CD schemes presented in our manuscript and in the supplementary material correspond to closed systems. In all cases, the CD grid behaves like the D grid as far as the propagation of the gravity waves is considered.

In short, we stand behind our linear analysis of the CD grid. Our conclusions are consistent with those of Skamarock (2008).

7-Further comments :

7a-In what sense are some schemes not closed?:

The CD grid discretization combines the C and D grid solutions in a time-split predictor-corrector sequence. The linearized system solves for the unknowns that are the values of predicted variables (divergence, vorticity, potential temperature etc.) in the next time step using the values of the known quantities that are provided for present time step.

The number of unknowns must be equal to the number of equations to have a closed system.

7b-What are the consequences if the system is not closed? :

With a scheme that is not closed, the solutions can include physically decoupled computational modes.

Here is a simple example: A large majority of the CD schemes that do not produce closed systems fail to produce a quadratic equation for the frequency, or an equivalent form given by Eq. (46a), which yields the same solution for positive or negative real-frequencies. Since the gravity waves horizontally propagate with the same speed every direction, this condition has to be satisfied by every "consistent" scheme.

Another example is the C-grid staggering on the triangular grid, which does not yield a closed system (Gassmann, 2011). An early version of ICON was based on a triangular grid and suffered from a checkerboard pattern its the divergence field as reported by its developers. Diffusion can of course render such noise invisible.

7c-How are these solutions affected by the use of high-order schemes? :

The use of high-order schemes has only a minor impact on our analysis. To see this, consider the discrete dispersion relation for the C-grid given by

[see Fig. 1 for this equation],

which is Eq. (17) of Part I. This form will be the same for high-order schemes, but with different definitions of ksi and eta. What is gained with the use of higher-order schemes is that the errors will be more confined near the SRHS. The errors at the SRHS are not improved by the use of higher-order schemes.

7d-Comments on the importance and use of linear analysis:

Linear analysis is an optimal tool to examine the behavior of waves near the SRHS. Obviously, we expect significant errors near the SRHS, but our manuscript demonstrates

that the nature and size of the errors depend on the grid used. In the development of a dynamical core, it is useful to choose the grid staggering that behaves as well as possible near the SRHS, all other factors being equal. If there are compelling reasons to select a different grid, then the advantages of that grid should be demonstrated through precise quantitative tests.

Linear tests also allow us to determine whether all of the primary dynamical processes properly interact with each other in the discrete system. In particular, the unknowns and the number of equations should be balanced for the system to be closed. Without such closure, uncontrolled modes may appear. The C-grid discretization of the momentum equations on the hexagonal system has been achieved in this way (Gassmann, 2011).

References

Skamarock, W. C. A linear analysis of the NCAR CCSM finite-volume dynamical core. Mon. Weather Rev., 136, 2112–2119, 2008:.

Gassmann, A.: Inspection of hexagonal and triangular C-grid discretizations of the shallow water equations. J. Comput. Phys., 230, 2706-2721, 2011.

[Figure]

$$v^2 = \frac{N^2\left(\xi^2 k^2 + \eta^2 \ell^2\right) + \mu^2 f^2 \left(m^2 + \dfrac{1}{4H^2}\right)}{\left(\xi^2 k^2 + \eta^2 \ell^2\right) + \left(m^2 + \dfrac{1}{4H^2}\right)}$$

**Fig. 1.** Equation

---

## Author Response (AR2)

[revised manuscript text omitted]
^2 = \frac{N^2(k^2+\ell^2)+f^2\left(m^2+\dfrac{1}{4H^2}\right)}{(k^2+\ell^2)+\left(m^2+\dfrac{1}{4H^2}\right)}$$

$$v^2 = \frac{N^2(\xi^2 k^2+\eta^2\ell^2)+f^2\left(m^2+\dfrac{1}{4H^2}\right)}{(\xi^2 k^2+\eta^2\ell^2)+\left(m^2+\dfrac{1}{4H^2}\right)} \quad 0 \le [kd,\,\ell d] \le \pi$$

$m \equiv \pi n/z_T$ for $n=1,2,3,\cdots$      $\xi \equiv \sin\left(\tfrac{1}{2}kd\right)/\left(\tfrac{1}{2}kd\right)$ and $\eta \equiv \sin\left(\tfrac{1}{2}\ell d\right)/\left(\tfrac{1}{2}\ell d\right)$

| C-grid | D-grid |
|---|---|

$$v^2 = \frac{N^2(\xi^2 k^2+\eta^2\ell^2)+\mu^2 f^2\left(m^2+\dfrac{1}{4H^2}\right)}{(\xi^2 k^2+\eta^2\ell^2)+\left(m^2+\dfrac{1}{4H^2}\right)}$$

$$v^2 = \frac{\mu^2 N^2(\xi^2 k^2+\eta^2\ell^2)+\mu^2 f^2\left(m^2+\dfrac{1}{4H^2}\right)}{\mu^2(\xi^2 k^2+\eta^2\ell^2)+\left(m^2+\dfrac{1}{4H^2}\right)}$$

$\mu \equiv \cos\left(\tfrac{1}{2}kd\right)\cos\left(\tfrac{1}{2}\ell d\right) \quad 0 \le [kd,\,\ell d] \le \pi$      $0 \le [kd,\,\ell d] \le \pi$

**CD-grid (Scheme I)**

$$e^{2v_r\tau}\left(\mu^2 L^2+\sigma_m^2\right)\cos(2v_r\tau) - 2e^{v_r\tau}\left(\mu^2\sigma_N L^2+\sigma_f\sigma_m^2\right)\cos(v_r\tau) \qquad e^{v_r\tau} = \frac{2\left(\mu^2\sigma_N L^2+\sigma_f\sigma_m^2\right)}{\left(\mu^2 L^2+\sigma_m^2\right)}\frac{\sin(v_r\tau)}{\sin(2v_r\tau)}$$

$$+\left(\sigma_N^2+\tau^2 N^2\right)\mu^2 L^2+\left(\sigma_f^2+\tau^2\mu^2 f^2\right)\sigma_m^2 = 0$$

$\sigma_N \equiv 1-\tfrac{1}{2}\tau^2 N^2 \quad \sigma_f \equiv 1-\tfrac{1}{2}\tau^2 f^2 \quad \sigma_m^2 \equiv m^2+1/\left(4H^2\right) \qquad 0 \le [kd,\,\ell d] \le \pi$

| A grid | E grid |
|---|---|

$$v^2 = \frac{N^2(\tilde{\xi}^2 k^2+\tilde{\eta}^2\ell^2)+f^2\left(m^2+\dfrac{1}{4H^2}\right)}{(\tilde{\xi}^2 k^2+\tilde{\eta}^2\ell^2)+\left(m^2+\dfrac{1}{4H^2}\right)} \quad 0 \le [kd,\,\ell d] \le \pi$$

$$v^2 = \frac{N^2(\xi^2 k^2+\eta^2\ell^2)+f^2\left(m^2+\dfrac{1}{4H^2}\right)}{(\xi^2 k^2+\eta^2\ell^2)+\left(m^2+\dfrac{1}{4H^2}\right)}$$

$\tilde{\xi} \equiv \sin(kd)/(kd)$ and $\tilde{\eta} \equiv \sin(\ell d)/(\ell d)$      $0 \le [kd,\,\ell d] \le 2\pi$

**B grid**

$$v^2 = \frac{N^2\left(\xi^2 k^2+\eta^2\ell^2-\dfrac{1}{2}d^2\xi^2 k^2\eta^2\ell^2\right)+f^2\left(m^2+\dfrac{1}{4H^2}\right)}{\left(\xi^2 k^2+\eta^2\ell^2-\dfrac{1}{2}d^2\xi^2 k^2\eta^2\ell^2\right)+\left(m^2+\dfrac{1}{4H^2}\right)} \quad 0 \le [kd,\,\ell d] \le \pi$$

| L grid | CP grid |
|---|---|

$$v^2 = \frac{\mu_z^2 N^2(k^2+\ell^2)+f^2\left(\zeta^2 m^2+\mu_z^2\dfrac{1}{4H^2}\right)}{(k^2+\ell^2)+\left(\zeta^2 m^2+\mu_z^2\dfrac{1}{4H^2}\right)}$$

$$v^2 = \frac{N^2(k^2+\ell^2)+f^2\left(\zeta^2 m^2+\mu_z^2\dfrac{1}{4H^2}\right)}{(k^2+\ell^2)+\left(\zeta^2 m^2+\mu_z^2\dfrac{1}{4H^2}\right)}$$

[revised manuscript text omitted]